# An Effective Levelling Paradigm for Unlabeled Scenarios

**Fangming Cui**[1,2]    **Zhou Yu**[3]    **Di Yang**[4]    **Yuqiang Ren**[4]
**Liang Xiao**[1]*    **Xinmei Tian**[5]*
[1]Defense Innovation Institute
[2]Shanghai Jiao Tong University    [3]The Key Laboratory of Complex Systems Modeling and
Simulation, the School of Computer Science, Hangzhou Dianzi University, China
[4]ByteDance Inc.    [5]MoE Key Laboratory of Brain-inspired Intelligent Perception and Cognition,
University of Science and Technology of China
cuifangming@sjtu.edu.cn yuz@hdu.edu.cn
xiaoliang@nudt.edu.cn xinmei@ustc.edu.cn

## Abstract

Advancements in direct-integration fine-tuning frameworks have underscored their potential to enhance the performance of labeled scenarios and tasks. To enhance the generalization of different categories in the same dataset, some methods have added visual loss to these frameworks for unlabeled scenarios. However, the performance of these methods through visual loss does not improve significantly in domain generalization and cross-dataset generalization tasks. This may be attributed to the uncoordinated learning of the two-modalities alignment and visual loss. To mitigate this issue of uncoordinated learning, we propose a novel method called Levelling Paradigm (LePa) to improve performance for unlabeled tasks or scenarios. The proposed LePa, designed as a plug-in module, dynamically constrains and coordinates multiple objective functions, thereby improving the generalization of these baseline methods. Comprehensive experiments have shown that our design can effectively address generalized scenarios and tasks.

## 1 Introduction

Vision-Language Models (VLMs), such as the frozen CLIP [1], are garnering increasing attention for their exceptional generalization capabilities. These VLMs have been trained to align textual and visual components by utilizing large datasets [2]. As an example, CLIP undergoes training on a massive dataset containing 400 million text-image pairs, enabling VLMs to encode a wide range of concepts in a unified embedding space. By aligning and integrating textual and visual information, VLMs can bridge the divide between language and visual representations, thereby improving their comprehension of context within the shared embedding space [3, 4]. VLMs and large language models [5, 6, 7] can exhibit strong performance across a diverse range of downstream tasks [8, 9, 10, 11]. One remarkable aspect of CLIP is its exceptional zero-shot generalization capability. This ability is achieved by employing predefined text inputs, like "a photo of a [class]," known as prompts, to generate recognition weights during the inference stage. Through the use of prompts, CLIP can effectively transfer its learned knowledge to achieve precise recognition, even for classes that have not been encountered before. The zero-shot capability enhances the adaptability and versatility of CLIP, allowing it to tackle various tasks [12, 13, 14] and domains without the need for extensive fine-tuning and adaptation [15].

---

*Corresponding authors.

Recent advancements involve keeping the weights of CLIP fixed while learning a set of textual parameters [16] for prompts to fine-tune CLIP for downstream image recognition tasks. Researchers have achieved notable advancements through investigations into the synchronization of images and prompts, surpassing the boundaries of earlier research endeavors. These analyses have acknowledged the promise of learning prompts for individual images, outstripping the efficacy of hand-crafted prompts. A pivotal revelation has been the importance of preserving the class name as inherent knowledge to guarantee that the acquired prompts can adeptly construct a classifier. Moreover, the word embeddings of prompts, serving as contextual indicators, are regarded as adjustable parameters. Remarkably, the trainable terms within the prompts are initialized using the "a photo of a" [1].

Recent research (Vision-Language Prompting, VLP) has found that initializing and fine-tuning visual prompt [17] directly based on learnable textual prompt can better improve the performance of supervised learning [18] and base classes of the same datasets [19, 20, 21]. In order to improve novel classes (unlabeled samples) within the same dataset, PromptSRC [22] incorporates traditional constraints to avoid forgetting general knowledge for enhancing the base-to-novel generalization through hand-crafted prompts (gray 'T' of Figure 1) and visual loss. Recently, CoPrompt [23] employs two learning adapters to enhance the base-to-novel generalization through the visual loss. ProMetaR [24] meta-learns both the visual regularizer and the soft prompts to harness the task-specific knowledge from the downstream tasks and task-agnostic general knowledge. However, the performance of these direct-integration VLP through visual loss [22, 23] does not improve significantly in higher difficulty generalization tasks, shown in Table 3 and Table 1 for unlabeled scenarios. This may be attributed to the uncoordinated learning [25, 26, 27] of the CE loss and visual loss [22, 23]. The theoretical basis is that the purpose of CE loss is to fine-tune and enhance few-shot learning ability [16], while the purpose of visual loss is to avoid forgetting pre-trained features and enhance generalization ability [28, 23, 24, 22]. There is a compromise and tradeoff between these two losses, a natural conflict where one goes up and the other goes down. To alleviate the uncoordinated learning, we design a flexible way to shape a dynamic constraint objective, improving the domain generalization and cross-dataset generalization. Our contributions can be summarized as follows:

- We propose a plug-and-play design called Levelling Paradigm (LePa) that is compatible with representative VLP baselines.
- The LePa dynamically coordinates multiple functions, thereby improving the generalization of direct-integration VLP with visual regularization.
- Representative tasks across 11 real datasets on generalization from base-to-novel, cross-dataset generalization, and domain generalization demonstrate that our design can effectively address generalized scenarios and tasks.

## 2 Methods

### 2.1 Frozen CLIP

Our method is founded on frozen CLIP [1] (as shown in Figure 1), a pioneering pre-trained vision-language model backbone known for its effectiveness in zero-shot learning applications. The pre-trained CLIP comprises two encoders: text encoder $\mathcal{G}_t(\cdot)$ and image encoder $\mathcal{G}_v(\cdot)$, which separately map textual input embeddings $\mathbf{p}$ and a visual training images, i.e., an image, $\mathbf{x}$ into a feature space through ViT blocks. The output features of two encoders of pre-trained CLIP are denoted as $\mathcal{G}_t(\mathbf{p})$ and $\mathcal{G}_v(\mathbf{x})$. Within the CLIP framework, the image encoder plays a crucial role in converting raw input images into feature embeddings, capturing intricate visual nuances, and deriving meaningful representations. Meanwhile, the text encoder is meticulously designed to generate representations for sequences of word embeddings, empowering the model to understand and process textual information effectively. Throughout the pre-training phase of CLIP, both the image and text encoders undergo simultaneous training on expansive datasets comprising text-image pairs. This alignment aligns seamlessly with the overarching goal of zero-shot recognition, a concept that can be formally delineated as follows,

$$p(z \mid \mathbf{x}) = \frac{\exp\left(\text{sim}\left(\mathcal{G}_t\left(\mathbf{p}_z\right), \mathcal{G}_v(\mathbf{x})\right)/\tau\right)}{\sum_{\mathbf{p}_i \in \mathcal{P}} \exp\left(\text{sim}\left(\mathcal{G}_t\left(\mathbf{p}_i\right), \mathcal{G}_v(\mathbf{x})\right)/\tau\right)}, \tag{1}$$

where $z$ is the label of training image $\mathbf{x}$, $\mathbf{p}_i$ denotes the pre-defined prompts, $\text{sim}(\cdot, \cdot)$ denotes the cosine similarity, and $\tau$ is temperature.

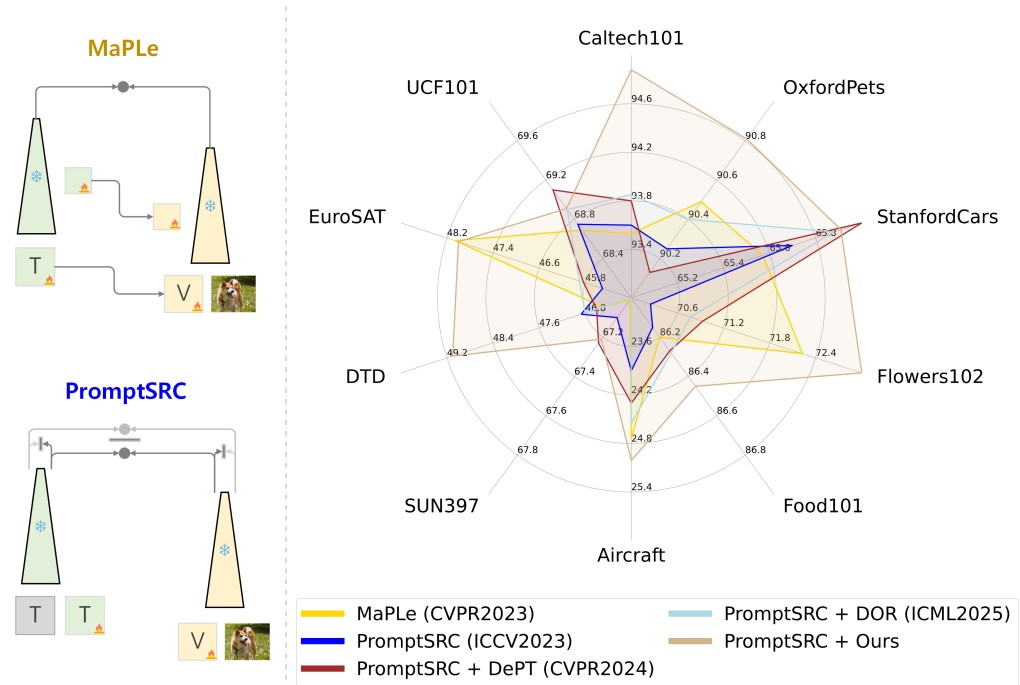

Figure 1: Performance comparison of Ours with prompting plug-in methods under the cross-dataset generalization. For cross-dataset generalization, the MaPLe (without visual loss) is higher than PromptSRC (with visual loss), as shown in Table 1. To improve the cross-dataset generalization of PromptSRC (with visual loss), we propose a plug-in design, improving the performance, surpassing the plug-and-play DePT [29] (CVPR2024) and DOR [30] (ICML2025). The green 'T' and yellow 'V' represent learnable textual prompts and visual prompts, respectively.

## 2.2 Language Prompt Learning

In recent studies, the weights of CLIP are kept unchanged [16], while a set of textual parameters are learned for prompts—these parameters help fine-tune CLIP for downstream image recognition. In this regard, textual prompting [31, 16, 32] has been proposed to improve base performance and few-shot supervised learning tasks. For class $c$, the tuning feature of the text encoder is denoted as $t_c$ in a dataset with total $C$ classes. The cross-entropy concept of the textual prompting method is computed as:

$$p(z \mid \mathbf{x}) = \frac{\exp\left(\text{sim}\left(t_c, \mathcal{G}_v(\mathbf{x})\right)/\tau\right)}{\sum_{c'=1}^{C} \exp\left(\text{sim}\left(t_{c'}, \mathcal{G}_v(\mathbf{x})\right)/\tau\right)}. \qquad (2)$$

## 2.3 Vision-Language Prompt Learning for Non-generalizable Fine-tuning

Recent research has found that initializing and learning visual prompt learning [17] directly based on textual prompt learning can better improve the performance of base classes of the same datasets and non-generalizable few-shot learning task [18]. Specifically, the output features of the text encoder in this Vison-Language Prompting (VLP) are denoted as $t_p$, and the output features of image encoder is denoted as $v_p$, where $\mathcal{L}_{\text{CE}}(\cdot, \cdot)$ is the cross-entropy loss with vision and language tuned prompts $\mathbf{p}$ for downstream training samples $\mathcal{N}$:

$$\mathcal{L}_{\text{CE}} = \arg\min_{\mathbf{p}} \mathbb{E}_{(\mathbf{x},z)\sim\mathcal{N}} \mathcal{L}\left(\text{sim}\left(t_p, v_p\right), z\right). \qquad (3)$$

Although VLP methods excel in base classes of base-to novel generalization, it still performs poorly in novel class recognition tasks (Avg. 11 datasets). This is because learnable prompts can overfit downstream data [22, 16].

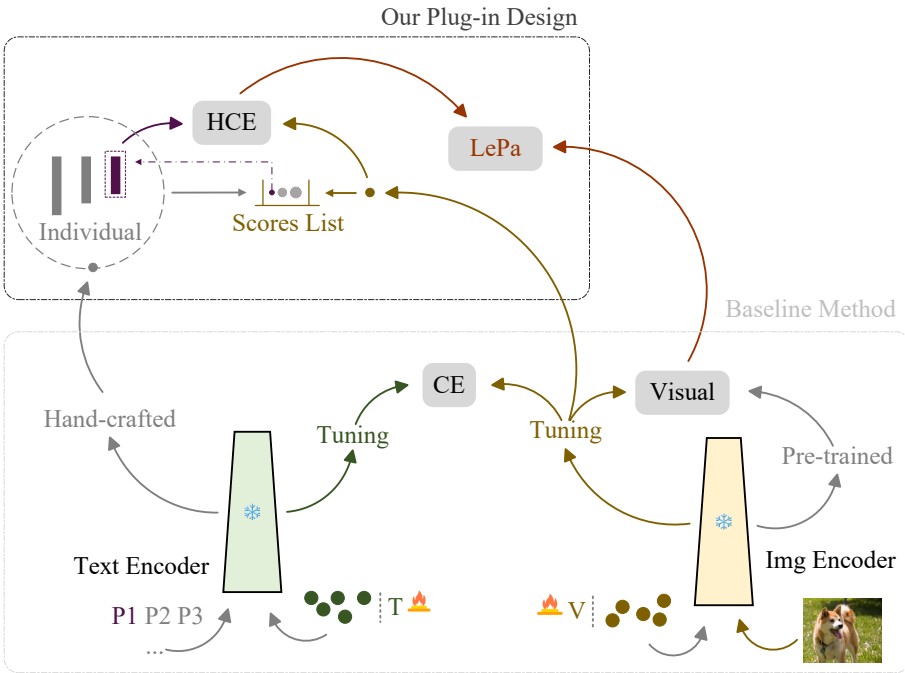

Figure 2: An overview of our plug-in design. We propose a plug-in design called Levelling Paradigm (LePa) in Figure 2 (Top). LePa regularizes the objective functions of two-modal alignment and visual regularization, thereby improving the robustness of coordinated fine-tuning and alleviating uncoordinated learning, thereby enhancing generalization tasks across 11 real datasets. Visual loss exists in baseline works [24, 23, 28, 22]. Three representative generalization experiments demonstrate that our method can effectively address generalized scenarios and tasks.

## 2.4 Visual Loss for Improving Generalizable Novel Classes

To improve this novel classes within the same dataset, some representative VLP works [22, 23, 28, 24] incorporates the constraint with $L1$ loss [33] on task-specific feature $v_p$ and pre-trained $v$ for visual branch, avoiding the forgetting of frozen CLIP's original generalization capability [34, 1, 22] and improving the novel classes of base-to-novel generalization task, as shown in Table 2. Following the $L1$ loss, the $\mathcal{L}_{\text{Visual}}(\cdot, \cdot)$ represents the visual loss:

$$\mathcal{L}_{\text{Visual}} = |v_p - v|. \tag{4}$$

## 2.5 Challenge and Motivation

However, the performance of these direct-integration VLP through visual loss does not improve significantly in domain generalization and cross-dataset generalization. For novel classes (Table 2), the MaPLe (without visual loss) is lower than PromptSRC (with visual loss). However, for cross-dataset generalization (Table 1), the MaPLe (without visual loss) is higher than PromptSRC (with visual loss). It seems that the visual loss performs poorly when faced with higher difficulty generalization tasks. This may be attributed to the uncoordinated learning [25, 26, 27] of the CE loss and visual loss. The theoretical basis is that the purpose of CE loss is to fine-tune and enhance few-shot learning ability [16], while the purpose of visual loss is to avoid forgetting pre-trained features and enhance generalization ability [22]. There is a compromise and tradeoff between these two losses, a natural conflict where one goes up and the other goes down.

## 2.6 Proposed Levelling Paradigm (LePa)

To improve the domain and cross-dataset generalizable performance and prevent visual loss convergence instability, we propose a novel plug-in method called Levelling Paradigm (LePa) that

normalizes Hand-crafted CE (HCE) and visual loss, using this method to maintain the stability of visual loss. HCE is composed of pre-trained manual text features, having generalization ability. Specifically, the LePa dynamically constrains the objective functions between vision-language alignment and visual regularization, thereby improving the robustness of coordinated fine-tuning. In Figure 2, these two objectives of $\mathcal{L}_{\text{CE}}$ and $\mathcal{L}_{\text{Visual}}$ may exhibit uncoordinated learning.

Specifically, uncoordinated learning can be defined as the scenario where the objective of the vision-language alignment performs well ($\mathcal{L}_{\text{CE}} \approx 0$) while the objective of visual regularization does not ($\mathcal{L}_{\text{Visual}} \gg 0$), or ($\mathcal{L}_{\text{CE}} \gg 0$) with ($\mathcal{L}_{\text{Visual}} \approx 0$). The difference in the value of these two objectives can serve as a measure of uncoordinated learning of direct-integration VLP with visual regularization. The uncoordinated learning not only diminishes the generalization learning of individual objectives but also disrupts the balance of objectives across the entire direct-integration VLP models.

Thus, we propose the $\mathcal{L}_{\text{LePa}}$ to balance the overall framework. We employ $N$ hand-crafted prompts (P1: a picture of a, P2: a photo of a, P3: a drawing of a, etc) to obtain individual $N$ hand-crafted features, as shown in Figure 2. We calculate cosine similarity as the scores between these hand-crafted generated features and the visual features generated by learnable visual embeddings of baseline methods. Afterwards, we will receive a score list.

Note that CLIP's general pre-trained features through hand-crafted prompts have a strong generalization ability [1, 22, 33]. In order to improve generalization, we fully capture the individual semantics of manual prompts in depth, and we obtained the hand-crafted prompt template with the $w$-worst score based on the scores list, and automatically recorded it. Further, we vectorized the recorded prompt word template text, and we average these $w$ manually output features to represent the overall situation of the bad case for robustness perspective. Further, we align the frozen hand-crafted textual features $\boldsymbol{t}$ with the learnable visual features $\boldsymbol{v}_p$ using cross-entropy loss to obtain $\mathcal{L}_{\text{HCE}}$.

In this regard, we employ the $\mathcal{L}_{\text{LePa}}$ to control the $\mathcal{L}_{\text{HCE}}$ and the visual loss $\mathcal{L}_{\text{Visual}}$ generated by pre-trained visual features $\boldsymbol{v}$ to alleviate uncoordinated learning of direct-integration VLP with visual regularization. Specifically, the $\mathcal{L}_{\text{LePa}}$ can be realized with the following form,

$$\left| \mathcal{L}_{\text{HCE}} \left( \boldsymbol{t}, \boldsymbol{v}_p \right) - \mathcal{L}_{\text{Visual}} \left( \boldsymbol{v}_p, \boldsymbol{v} \right) \right|. \tag{5}$$

Here, model parameters are omitted for simplicity, $\boldsymbol{v}_p$ represents the task-specific visual embeddings. According to the equation, minimizing the objective function $\mathcal{L}_{\text{LePa}}$ can make the model calibrate the uncoordinated learning of entire framework, reflecting in the performance improvement of higher difficulty generalization experiments (Table 1). Based on the above design, we can alleviate the situation of hindering model tuning and alignment: V-L alignment performs well, with the visual branch does not. The training objective of our method involves hyperparameters $\gamma_1$ and $\gamma_2$. The total loss $\mathcal{L}_{\text{total}}$ can be formulated as follows,

$$\mathcal{L}_{\text{total}} = \mathcal{L}_{\text{CE}} + \gamma_1 \mathcal{L}_{\text{LePa}} + \gamma_2 \mathcal{L}_{\text{Visual}}. \tag{6}$$

The proposed design highlights the necessity for generating features that ensure consistent performance across various objectives. Recognizing and addressing this uncoordinated learning would aid in promoting uniformity through the optimization of embeddings and models, aligning with the ethos of bridging the divide between language and visual representations of CLIP. Consequently, LePa regularizes the interaction between two-modal alignment and visual regularization to enhance the performance of the existing direct-integration approach, thereby addressing the uncoordinated learning in these frameworks [22, 23, 28, 24]. It ensures continuous robustness of the overall direct-integration VLP with visual regularization. Note that the proposed LePa is a plug-in design that is compatible with existing direct-integration VLP frameworks, as shown in (Table 3, Table 1, and Table 2).

## 3 Main Results

We introduce compare methods, implementation details, 11 datasets, and the key experiments (cross-dataset generalization, domain generalization, and base-to-novel generalization).

### 3.1 Compared Prompting Methods

We conduct performance analysis based on various direct-integration VLP baselines, including MaPLe [19] (CVPR2023), PromptSRC (PSRC) [22] (ICCV2023), and CoPrompt (CoP) [23]

(ICLR2024). The MaPLe employs the learnable hidden multi-layer for the fusion of two encoders without visual loss. The PromptSRC is independent vision and language prompts which employs traditional KL loss and L1 loss to avoid forgetting pre-trained knowledge through visual loss. The CoPrompt exists learnable hidden multi-layer which employs two learning adapters and two perturbed inputs to enhance the base-to-novel generalization through visual loss. The DePT [29] (CVPR2024) and DOR [30] (ICML2025) is plug-in methods. For the compared prompting methods, we adopt the optimal settings. We use a ViT-B/16-based CLIP model with compared methods for fair comparison.

Table 1: Cross-dataset generalization. * indicates our reproduced results.

| | Source | Target | | | | | | | | | | |
| | ImageNet | Caltech101 | OxfordPets | StanfordCars | Flowers102 | Food101 | Aircraft | SUN397 | DTD | EuroSAT | UCF101 | Average |
|---|---|---|---|---|---|---|---|---|---|---|---|---|
| MaPLe | 70.72 | 93.53 | 90.49 | 65.57 | 72.23 | 86.20 | 24.74 | 67.01 | 46.49 | 48.06 | 68.69 | 66.30 |
| + DePT | 73.27 | 92.55 | 90.22 | 64.58 | 70.09 | 85.33 | 23.78 | 66.26 | 45.11 | 40.25 | 67.88 | 64.60 |
| + DOR | 71.50 | 93.10 | 90.00 | 64.11 | 73.23 | 86.64 | 25.13 | 67.04 | 46.25 | 49.30 | 68.70 | 66.35 |
| + Ours | 72.33 | 94.11 | 90.85 | 65.88 | 73.81 | 86.55 | 24.81 | 68.89 | 46.73 | 49.22 | 68.11 | **66.89** |
| PromptSRC | 71.27 | 93.60 | 90.25 | 65.70 | 70.25 | 86.15 | 23.90 | 67.10 | 46.87 | 45.50 | 68.75 | 65.81 |
| + DePT | 71.60 | 93.80 | 90.13 | 66.00 | 70.93 | 86.27 | 24.30 | 67.23 | 46.60 | 45.83 | 69.10 | 66.02 |
| + DOR | 71.55 | 93.85 | 90.40 | 65.88 | 70.77 | 86.28 | 24.55 | 67.00 | 46.80 | 45.90 | 68.90 | 66.03 |
| + Ours | 71.50 | 94.88 | 90.81 | 65.91 | 73.00 | 86.45 | 25.01 | 67.21 | 49.10 | 48.00 | 68.91 | **66.93** |
| CoPrompt* | 70.70 | 93.40 | 90.15 | 65.38 | 72.15 | 86.00 | 24.10 | 66.08 | 47.25 | 51.15 | 69.00 | 65.46 |
| + DePT | 71.01 | 93.50 | 90.18 | 66.33 | 70.71 | 86.00 | 24.10 | 67.08 | 46.01 | 45.06 | 69.40 | 65.84 |
| + DOR | 71.00 | 93.11 | 90.03 | 66.04 | 70.32 | 86.10 | 24.23 | 67.13 | 45.71 | 45.20 | 69.00 | 65.69 |
| + Ours | 71.08 | 93.15 | 90.30 | 66.33 | 73.01 | 87.88 | 24.62 | 67.00 | 46.96 | 48.05 | 69.10 | **66.64** |

## 3.2 Implementation Details

After the ablation experiment, we obtained the implementation details. We set textual and visual embeddings to 4 based on all VLP methods. We use deep prompting with multi-modal encoders and an SGD optimizer with a learning rate of 0.0026 on a single A5000 GPU. Training for 30 epochs for base-to-novel generalization by 16-shot, 20 epochs for domain generalization and cross-dataset evaluation setting. We train the ImageNet source model on all classes with 16-shot in the first 3 transformer layers for domain generalization and cross-dataset evaluation. We set $\gamma_1 = \gamma_2 = 5$ for multi-modal regularization in total loss. We set $w = 4$ for $w$-worst cases. For the base-to-novel generalization, we set the learning depth to 9. We fix $N = 60$ hand-crafted prompts [1, 22], following CLIP. The hand-crafted prompt template used in this paper is: "a photo of a .", "a bad photo of a .", "a photo of many .", "a sculpture of a .", "a photo of the hard to see .", etc.

## 3.3 Datasets

The open-source real datasets cover multiple recognition tasks. We conducted base-to-novel generalization experiments and cross-dataset generalization experiments on 11 datasets. We conduct domain generalization experiments on four variants of ImageNet [35]. The datasets encompass various recognition tasks, including ImageNet [35], Caltech101 [36] for generic objects, OxfordPets [37], StanfordCars [38], Flowers102 [39], Food101 [40], FGVCAircraft [41] for fine-grained classification, SUN397 [42] for scene recognition, UCF101 [43] for action recognition, DTD [44] for texture classification, and EuroSAT [45] for satellite images. For the domain generalization benchmark, we use ImageNetA [46], ImageNet-R [47], ImageNet-Sketch [48] and ImageNetV2 [49].

## 3.4 Cross-Dataset Generalization Task

In Table 1, training on the source dataset (first column) is a non-generalization task only for ImageNet. The model trained on ImageNet will directly obtain an accuracy of 50,000 images, which is the source data in the first column of the Table 1. We test our ImageNet-trained model directly on the other 10 datasets to validate the potential of our method in cross-dataset transfer [50]. This experimental

Table 2: Base-to-novel generalization. * indicates our reproduced results.

(a) **Avg. 11 datasets**

|  | Base | Novel | HM |
|---|---|---|---|
| MaPLe | 82.28 | 75.14 | 78.55 |
| + DePT | 84.85 | 74.82 | 79.52 |
| + DOR | 84.13 | 76.15 | 79.94 |
| + Ours | 84.48 | 76.46 | **80.27** |
| PSRC | 84.26 | 76.10 | 79.97 |
| + DePT | 85.19 | 76.17 | 80.43 |
| + DOR | 84.34 | 76.99 | 80.48 |
| + Ours | 85.61 | 77.46 | **81.33** |
| CoP* | 83.89 | 76.75 | 80.13 |
| + DePT | 84.03 | 76.77 | 80.23 |
| + DOR | 84.10 | 77.69 | 80.76 |
| + Ours | 84.86 | 77.79 | **81.16** |

(b) ImageNet

|  | Base | Novel | HM |
|---|---|---|---|
| MaPLe | 76.66 | 70.54 | 73.47 |
| + DePT | 77.87 | 70.23 | 73.85 |
| + DOR | 76.57 | 72.00 | 74.23 |
| + Ours | 78.03 | 71.05 | 74.37 |
| PSRC | 77.60 | 70.73 | 74.01 |
| + DePT | 78.20 | 70.27 | 74.02 |
| + DOR | 77.71 | 71.54 | 74.49 |
| + Ours | 78.81 | 71.44 | 74.94 |
| CoP* | 77.00 | 71.10 | 74.02 |
| + DePT | 77.01 | 71.05 | 74.00 |
| + DOR | 77.21 | 72.03 | 74.58 |
| + Ours | 78.15 | 70.88 | 74.46 |

(c) Caltech101

|  | Base | Novel | HM |
|---|---|---|---|
| MaPLe | 97.74 | 94.36 | 96.02 |
| + DePT | 98.53 | 95.03 | 96.75 |
| + DOR | 97.55 | 94.90 | 96.22 |
| + Ours | 98.28 | 96.00 | 97.12 |
| PSRC | 98.10 | 94.03 | 96.02 |
| + DePT | 98.57 | 94.10 | 96.28 |
| + DOR | 98.25 | 94.80 | 96.51 |
| + Ours | 98.55 | 95.91 | 97.21 |
| CoP* | 97.30 | 95.05 | 96.17 |
| + DePT | 97.81 | 95.11 | 96.46 |
| + DOR | 97.44 | 96.00 | 96.71 |
| + Ours | 98.00 | 95.10 | 96.54 |

(d) OxfordPets

|  | Base | Novel | HM |
|---|---|---|---|
| MaPLe | 95.43 | 97.76 | 96.58 |
| + DePT | 95.03 | 97.83 | 96.41 |
| + DOR | 95.55 | 98.50 | 97.02 |
| + Ours | 95.76 | 98.50 | 97.11 |
| PSRC | 95.33 | 97.30 | 96.30 |
| + DePT | 95.43 | 97.33 | 96.37 |
| + DOR | 95.66 | 98.40 | 97.03 |
| + Ours | 95.61 | 97.80 | 96.69 |
| CoP* | 95.10 | 97.13 | 96.11 |
| + DePT | 94.60 | 97.35 | 95.97 |
| + DOR | 95.35 | 98.00 | 96.67 |
| + Ours | 95.01 | 98.13 | 96.56 |

(e) EuroSAT

|  | Base | Novel | HM |
|---|---|---|---|
| MaPLe | 94.07 | 73.23 | 82.35 |
| + DePT | 94.43 | 76.23 | 84.36 |
| + DOR | 94.11 | 74.25 | 82.84 |
| + Ours | 94.19 | 75.71 | 83.94 |
| PSRC | 92.90 | 73.90 | 82.32 |
| + DePT | 92.23 | 77.90 | 84.88 |
| + DOR | 92.80 | 74.81 | 82.71 |
| + Ours | 95.41 | 76.90 | 85.16 |
| CoP* | 94.24 | 75.88 | 84.12 |
| + DePT | 94.75 | 76.00 | 84.59 |
| + DOR | 94.56 | 76.60 | 84.94 |
| + Ours | 95.11 | 76.32 | 84.97 |

(f) UCF101

|  | Base | Novel | HM |
|---|---|---|---|
| MaPLe | 83.00 | 78.66 | 80.77 |
| + DePT | 86.87 | 78.10 | 82.25 |
| + DOR | 83.22 | 79.60 | 81.38 |
| + Ours | 87.00 | 79.31 | 82.97 |
| PSRC | 87.10 | 78.80 | 82.74 |
| + DePT | 87.73 | 77.70 | 82.46 |
| + DOR | 87.21 | 79.60 | 83.32 |
| + Ours | 88.74 | 79.11 | 83.65 |
| CoP* | 86.00 | 79.63 | 82.76 |
| + DePT | 86.10 | 79.80 | 82.93 |
| + DOR | 86.44 | 80.77 | 83.56 |
| + Ours | 86.81 | 80.00 | 83.37 |

(g) StanfordCars

|  | Base | Novel | HM |
|---|---|---|---|
| MaPLe | 72.94 | 74.00 | 73.47 |
| + DePT | 80.93 | 71.73 | 76.06 |
| + DOR | 92.87 | 75.10 | 83.02 |
| + Ours | 78.51 | 75.05 | 76.74 |
| PSRC | 78.27 | 74.97 | 76.58 |
| + DePT | 80.80 | 75.00 | 77.79 |
| + DOR | 78.50 | 75.90 | 77.19 |
| + Ours | 80.80 | 76.70 | 78.69 |
| CoP* | 80.80 | 74.33 | 77.50 |
| + DePT | 81.13 | 74.00 | 77.49 |
| + DOR | 80.95 | 75.50 | 78.19 |
| + Ours | 81.12 | 76.05 | 78.56 |

(h) Flowers102

|  | Base | Novel | HM |
|---|---|---|---|
| MaPLe | 95.92 | 72.46 | 82.56 |
| + DePT | 98.03 | 73.17 | 83.79 |
| + DOR | 95.80 | 74.00 | 83.63 |
| + Ours | 98.18 | 74.88 | 84.96 |
| PSRC | 98.07 | 76.50 | 85.95 |
| + DePT | 98.40 | 77.10 | 86.46 |
| + DOR | 98.10 | 77.40 | 86.89 |
| + Ours | 98.70 | 78.36 | 87.36 |
| CoP* | 96.10 | 77.25 | 86.04 |
| + DePT | 96.81 | 77.05 | 86.33 |
| + DOR | 96.65 | 77.94 | 86.59 |
| + Ours | 97.88 | 78.02 | 87.27 |

(i) Food101

|  | Base | Novel | HM |
|---|---|---|---|
| MaPLe | 90.71 | 92.05 | 91.38 |
| + DePT | 90.33 | 91.53 | 90.93 |
| + DOR | 90.88 | 92.90 | 91.88 |
| + Ours | 90.28 | 92.95 | 91.59 |
| PSRC | 90.67 | 91.53 | 91.10 |
| + DePT | 90.87 | 91.57 | 91.22 |
| + DOR | 90.90 | 92.63 | 91.76 |
| + Ours | 90.58 | 92.51 | 91.53 |
| CoP* | 90.65 | 92.03 | 91.34 |
| + DePT | 91.00 | 92.72 | 91.85 |
| + DOR | 90.45 | 93.00 | 91.70 |
| + Ours | 91.10 | 92.57 | 91.83 |

(j) FGVCAircraft

|  | Base | Novel | HM |
|---|---|---|---|
| MaPLe | 37.44 | 35.61 | 36.50 |
| + DePT | 44.53 | 32.80 | 37.78 |
| + DOR | 37.85 | 36.70 | 37.32 |
| + Ours | 43.00 | 37.35 | 39.97 |
| PSRC | 42.73 | 37.87 | 40.15 |
| + DePT | 45.70 | 36.73 | 40.73 |
| + DOR | 42.80 | 38.71 | 40.65 |
| + Ours | 45.78 | 39.87 | 42.62 |
| CoP* | 41.40 | 39.80 | 40.58 |
| + DePT | 41.01 | 38.75 | 39.85 |
| + DOR | 41.51 | 40.60 | 41.05 |
| + Ours | 45.12 | 44.20 | 44.65 |

(k) SUN397

|  | Base | Novel | HM |
|---|---|---|---|
| MaPLe | 80.82 | 78.70 | 79.75 |
| + DePT | 82.90 | 76.40 | 79.52 |
| + DOR | 80.76 | 79.56 | 80.16 |
| + Ours | 82.13 | 79.65 | 80.87 |
| PSRC | 82.67 | 78.47 | 80.52 |
| + DePT | 83.27 | 78.97 | 81.06 |
| + DOR | 82.80 | 79.35 | 81.04 |
| + Ours | 83.88 | 80.77 | 82.29 |
| CoP* | 82.49 | 78.70 | 80.56 |
| + DePT | 82.16 | 79.10 | 80.59 |
| + DOR | 82.68 | 80.01 | 81.32 |
| + Ours | 83.01 | 80.11 | 81.54 |

(l) DTD

|  | Base | Novel | HM |
|---|---|---|---|
| MaPLe | 80.36 | 59.18 | 68.16 |
| + DePT | 83.87 | 59.93 | 69.91 |
| + DOR | 80.35 | 60.20 | 68.84 |
| + Ours | 83.92 | 60.65 | 70.41 |
| PSRC | 83.37 | 62.97 | 71.75 |
| + DePT | 84.80 | 61.20 | 71.09 |
| + DOR | 83.11 | 63.81 | 72.19 |
| + Ours | 84.90 | 62.70 | 72.13 |
| CoP* | 81.78 | 63.40 | 71.45 |
| + DePT | 82.06 | 63.55 | 71.63 |
| + DOR | 81.90 | 64.22 | 71.99 |
| + Ours | 82.25 | 64.39 | 72.26 |

setup is similar to the domain generalization experimental setup. Compared with DePT, our method shows improved performance in average precision. For cross-dataset generalization, the MaPLe (without visual loss) is higher than PromptSRC (with visual loss). Through our plugin integration, the cross-dataset performance of PromptSRC has been improved. In Figure 1, it suggests that our method demonstrates a clear advantage in terms of generalization across most classification scenes.

### 3.5 Base-to-Novel Generalization Task

This experiment is to test whether the model can handle small-scale generalization. We follow a setting where the same datasets are split into base and novel classes. So the distribution of the base classes is similar to that of the novel classes. Dividing all classes of a dataset into two parts is a process of random division. Please note that this is divided into two equally sized parts. The model is trained only on the base classes in a 16-shot setting and tested on base (non-generalization task) and novel classes (generalization task). Finally, we use the HM (Harmonic Mean) [51] to evaluate the model's ability to balance generalization and non-generalization. The HM task aims to observe the model's fine-tuning ability and similar classes classification ability, expressed as:

Table 3: Domain generalization Task. * indicates our reproduced results.

| | Source | Target | | | |
|---|---|---|---|---|---|
| | ImageNet | -V2 | -S | -A | -R |
| MaPLe | 70.72 | 64.07 | 49.15 | 50.90 | 76.98 |
| + DePT | 73.27 | 65.00 | 49.05 | 51.15 | 77.30 |
| + DOR | 71.50 | 64.94 | 48.56 | 52.00 | 77.10 |
| + Ours | 72.33 | 65.41 | 50.40 | 52.15 | 78.15 |
| PSRC | 71.27 | 64.35 | 49.55 | 50.90 | 77.80 |
| + DePT | 71.60 | 64.51 | 50.15 | 51.88 | 77.18 |
| + DOR | 71.55 | 64.00 | 50.20 | 52.15 | 77.65 |
| + Ours | 71.50 | 65.15 | 51.05 | 52.33 | 78.75 |
| CoP* | 70.70 | 64.15 | 49.48 | 50.60 | 77.40 |
| + DePT | 71.01 | 64.60 | 50.05 | 51.05 | 77.45 |
| + DOR | 71.00 | 63.90 | 50.13 | 52.00 | 77.51 |
| + Ours | 71.08 | 65.60 | 51.08 | 52.00 | 78.20 |

$$HM = \frac{2 \times Acc_{\text{base}} \times Acc_{\text{novel}}}{Acc_{\text{base}} + Acc_{\text{novel}}}. \qquad (7)$$

As shown in Table 2 (left-top), our plugin design is based on these representative VLP architectures, which have all improved novel performance. Moreover, previous methods have compromised their generalization capabilities when it comes to handling more specialized datasets. For the specialized EuroSAT [45], our method provides the highest novel accuracy. In Table 2, our method provides the best-averaged results on the novel classes. Overall, our method demonstrates significant improvements on an average of 11 datasets for HM.

### 3.6 Domain Generalization Task

In Table 3, we train our method on the ImageNet dataset with 1,000 classes and then test it on the domain shift datasets to evaluate the robustness [52, 53, 54] of our approach. In domain generalization experiments, training on the source dataset, ImageNet, is a non-generalization task only for this dataset. The remaining 4 ImageNet variant datasets are tested directly using ImageNet-trained models to determine the model's generalization ability. Our method consistently outperforms all existing baselines (MaPLe, PromptSRC, CoPrompt) on target datasets. When compared to plug-in DePT [29] and DOR [30], our method demonstrates improved performance across variants of the ImageNet dataset. Our method is purposefully crafted to bolster generalization capabilities when faced with domain shifts. This experiment is commonly set up in real-life scenarios, such as whether the model can recognize objects with minor tampering.

## 4 Further Analysis

In this section, we conduct crucial ablation experiments. The first experiment is cost training, and the second experiment is cross-dataset generalization experiment with different semantics.

### 4.1 Computational Cost

In Table 4a and Table 4b, the computational cost analysis is performed using the SUN397 over 10 epochs on a single GPU. Our method (row 3) may exhibit more training time due to the necessity of performing multiple cosine similarity calculations. In future work, we will consider how to alleviate

this time consumption. Compared to PromptSRC and MaPLe, our plug-and-play method does not increase in terms of parameter count. The DePT method increases the number of learning parameters, which are related to the parameters of the base class. Our component only adds the type of loss.

Table 4: The cost analysis is performed using the SUN397 [42] over 10 epochs. 'N': the number of classes in the base task [29].

(a) The cost analysis is based on PromptSRC [22].

| Method | Train time | Learnable para. | HM |
|---|---|---|---|
| PromptSRC | 13.13 min | 5120 | 79.97 |
| + DePT | 13.88 min | + (2+N/2) K | 80.13 |
| + Ours | 14.80 min | + 0 K | 81.01 |

(b) The cost analysis is based on MaPLe [19].

| Method | Train time | Learnable para. | HM |
|---|---|---|---|
| MaPLe | 10.55 min | 3.55M | 79.59 |
| + DePT | 10.66 min | + (2+N/2)K | 79.69 |
| + Ours | 13.56 min | + 0 K | 79.98 |

## 4.2 Analysis of Semantic Misalignment

In Table 5, we design and conduct a new experimental setup. In this experiment, we trained 10 datasets (Caltech101, OxfordPets, StanfordCars, Flowers102, Food101, FGVCAircraft, SUN397, UCF101, DTD, EuroSAT) with with 16-shot learning and 5 epochs, and then tested each dataset directly on the remaining 9 datasets. We did not use the ImageNet dataset because it has too many categories and would include other datasets. This setting is to simulate the situation of semantic mismatch. For example, after the model is trained on the SUN397 dataset, the current embeddings are fitted to the SUN397 dataset. Then when we use this model to test the Food101 dataset, the semantics of the text branch embeddings and the visual branch test dataset are not aligned. The cross-dataset generalization of this pattern is a difficult generalization setting, and in real-world scenarios, the distribution gap between the trained dataset and the generalized dataset is often very large. We use this experiment to observe the generalization effect of this pattern in different unlabeled scenarios.

Table 5: Analysis of Semantic Misalignment.

| Source | Target | | | | | | | | | |
|---|---|---|---|---|---|---|---|---|---|---|
| | Caltech101 [36] | Food101 [40] | DTD [44] | UCF101 [43] | Flowers102 [39] | OxfordPets [37] | Aircraft [41] | StanfordCars [38] | SUN397 [42] | EuroSAT [45] |
| Caltech101 (94.7) | | 86.0 | 45.6 | 65.2 | 67.2 | 88.3 | 23.8 | 65.5 | 65.7 | 44.6 |
| Food101 (87.1) | 92.7 | | 43.3 | 64.9 | 67.9 | 88.3 | 23.5 | 65.5 | 64.8 | 47.5 |
| DTD (58.3) | 93.1 | 85.6 | | 64.8 | 69.8 | 87.5 | 22.4 | 64.9 | 62.0 | 43.7 |
| UCF101 (76.5) | 91.0 | 85.4 | 43.7 | | 65.9 | 83.1 | 20.9 | 63.7 | 62.8 | 42.8 |
| Flowers102 (87.5) | 89.5 | 80.5 | 39.1 | 58.2 | | 82.9 | 19.9 | 44.6 | 52.6 | 51.9 |
| OxfordPets (93.1) | 93.1 | 86.2 | 42.1 | 63.5 | 66.4 | | 21.6 | 44.6 | 52.6 | 48.7 |
| Aircraft (31.0) | 81.3 | 84.1 | 40.7 | 63.3 | 62.4 | 83.2 | | 44.6 | 58.2 | 45.0 |
| StanfordCars (69.2) | 90.3 | 85.2 | 40.7 | 62.8 | 59.4 | 85.0 | 22.3 | | 61.0 | 47.7 |
| SUN397 (70.0) | 91.9 | 85.7 | 38.4 | 63.9 | 63.7 | 83.2 | 21.7 | 62.4 | | 49.1 |
| EuroSAT (76.7) | 84.3 | 68.7 | 31.2 | 54.2 | 32.0 | 43.6 | 17.5 | 39.1 | 53.6 | |

# 5 Related Work

## 5.1 Vision-Language Models (VLMs)

VLMs have revolutionized the field of artificial intelligence by seamlessly blending visual and textual data to enhance comprehension of multimodal information [55, 56, 57]. This innovative technology serves as a robust solution for analyzing and producing content that merges images and text, resulting in major advancements across diverse domains like image captioning [4, 58, 59] and other tasks [60, 61, 62]. Central to the architecture of VLMs is the amalgamation of visual and textual modalities, enabling the system to acquire intricate representations that encapsulate the semantic

connections between images and their corresponding textual descriptions [63, 64]. Through the concurrent processing of visual and textual data, VLMs effectively bridge the divide between disparate modalities, generating coherent outputs that capitalize on the synergistic information inherent in each domain [65, 66, 67]. The VLMs can also demonstrate value in other fields [68, 69, 70].

## 5.2 Prompt Learning

Prompt Learning serves as a prevalent method in the field of NLP, aiding in the acquisition of skills for various subsequent tasks [71, 72]. Employing textual prompts [73], serving as directives for the linguistic component of a VLMs, replicating this practice is a prevalent method to augment comprehension of tasks. The adapter [74, 75] approaches can achieve competitive performance to adapt VLMs to downstream tasks [76, 77, 78]. Nevertheless, the limitations of these approaches have spurred the investigation of novel techniques inspired by prompt tuning in Natural Language Processing (NLP). The image-conditional prompt [31] significantly contributes to enhancing generalization to unseen classes (unlabeled samples). This conditioning aids in improving the generalization to unseen examples. Some methods [32, 33] constrain the learnable prompts to prior distribution learning. TAP [79] first instructs large language models to generate a tree of attributes with a "concept - attribute - description" structure for each category. In addition to single-modal prompt tuning and two-modal prompt tuning, FAP [80] introduces robust attack [81, 82, 83] for prompt learning based on VLP. Moreover, prompt learning can also demonstrate value in robot fields [84, 85] and innovative research areas [86, 87].

## 6 Limitations

Our design has more training time, we will explore how to reduce this overhead. In Table 3 (column-1) and Table 1 (column-1), it seems that our method has a lower performance compared to DePT on the source dataset of training labeled ImageNet. The LePa aims to enhance cross-dataset generalization and domain generalization experiments, but there is a compromise and tradeoff with specifying ImageNet dataset fine-tuning. In real-world industrial settings, obtaining labeled training data can often be challenging or impractical. In future work, we will try to improve our performance on few-shot learning and generalization, such as 1-shot and 2-shot scenarios with little training data.

## 7 Conclusion

It has recently been discovered that VLP frameworks have underscored their potential to improve the fine-tuning performance of labeled scenarios for pre-trained CLIP. In order to improve the generalization of novel classes in the same dataset, some methods have added visual regularization to VLP frameworks. However, the performance of these baseline methods does not improve significantly in domain generalization and cross-dataset generalization tasks. This may be attributed to the uncoordinated learning of the fine-tuning CE loss and generalizable visual loss. To address this problem of uncoordinated learning, we propose an effective design called Levelling Paradigm (LePa) to improve performance for unlabeled tasks or scenarios. The proposed LePa, designed as a plug-and-play method, dynamically constrains and coordinates multiple objective functions, thereby enhancing the generalization of these baseline methods. Representative tasks across 11 real datasets on generalization from base-to-novel, cross-dataset generalization, and domain generalization demonstrate that our design can effectively address generalized scenarios and tasks.

**Broader Impacts.** This paper presents work whose goal is to advance the field of Machine Learning. None of these points we feel must be specifically highlighted here. This finding carries important implications for the deployment of VLMs across diverse real-world applications. By enhancing zero-shot recognition capabilities, our approach offers substantial benefits to industries that depend on large-scale image analysis, including the improvement of visual search systems, the refinement of automated image annotation pipelines, and the advancement of tools in imaging. The societal implications of our research involve democratizing the availability of potent AI resources, as our approach can achieve impressive performance even in the absence of extensive labeled data, thereby enhancing the accessibility and utility of advanced VLMs in environments with limited resources. Furthermore, our method promotes ethical AI practices by minimizing the necessity for extensive model training and adaptation, thereby supporting sustainability and efficiency objectives.

## Acknowledgement

We extend our heartfelt gratitude to the anonymous reviewers for their insightful comments, which greatly improved the quality of this paper. We sincerely express our gratitude to the anonymous AC for the responsible coordination throughout the entire process. This work was supported in part by NSFC No. 62222117. This work was supported in part by the National Natural Science Foundation of China under Grants 62422204, and in part by the Zhejiang Provincial Natural Science Foundation of China under Grant LDT23F02025F02, and in part by the Key Research and Development Program of Zhejiang Province under Grant 2025C01026.

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
