# OpenReview forum: "An Effective Levelling Paradigm for Unlabeled Scenarios"
_NeurIPS.cc/2025/Conference — NeurIPS 2025 poster_

### Official Review · Reviewer_w5x5 · 2025-06-02

**Clarity:** 2
**Significance:** 3
**Originality:** 3
**Rating:** 4
**Confidence:** 2

**Summary:**

This paper proposes a prompt tuning for VLM on unlabeled tasks or scenarios. PromptSrc or FAP has improved performance on labeled tasks using VLM like CLIP, but they often underperform on unlabeled tasks, particularly in domain generalization and cross-dataset generalization. This shortfall is attributed to uncoordinated learning of textual and visual embeddings in these methods. This paper proposes a plug-in method (LePa) that coordinates the optimization of multiple objectives (in text and visual modalities). LePa shapes the loss landscape to improve robustness during fine-tuning, especially in unlabeled or domain-shifted scenarios. Existing VLP designs align text and vision modalities separately, which might cause visual feature tuning (learned from image encoders) to fail or diverge, leading to unbalanced or degraded generalization, especially in domain generalization and cross-dataset tasks. It outperforms several baselines on PromptSRC/FAP/TAP/MaPLe on 11 datasets.

**Questions:**

See above.

**Ethical Concerns:**

["NO or VERY MINOR ethics concerns only"]

**Limitations:**

Yes.

**Paper Formatting Concerns:**

No.

**Quality:**

3

**Strengths And Weaknesses:**

LePa introduces a lightweight method to regularize and balance learning in vision-language prompting by aligning the objectives of text-to-image alignment and visual tuning. It is compatible with 4 existing methods on vlm prompting and effective in 11 datasets (mostly) but may suffer from limitations in scalability, hyperparameter tuning and manual design dependecies. In terms of scalability, LePA requires manual prompts and handcrafted feature vectors to initialize score lists. This adds preprocessing overhead and may not scale to large prompt spaces or tasks without well-defined templates. Moreover, ablation on each term $\gamma_{1}$ and $\gamma_{2}$ are important (Table 5a) only presents $\gamma_{1}$ and $\gamma_{2}$ are positive scenarios. It has several interpretability issues as the calibration logic is somewhat ad hoc; it relies on choosing "worst" prompts and averaging their performance, which may be sensitive to prompt variability or batch statistics.Assumes that minimizing the alignment gap between $L_{HCE}$ and $L_{visual}$ improves generalization. This may not hold if both branches are misaligned for fundamental reasons (e.g., semantic mismatch). Unlabeled tasks might contain semantic mismatch and did the authors perform experiments on samples in unlabeled tasks that contain semantic mismatch data?

---

> ### Author Rebuttal · Authors · 2025-07-29
>
> # ***Thanks to the reviewer:***
>
> > We thank the reviewers for their evaluation of our current manuscript. We understand that there are shortcomings in our work and we will **continue to in-depth research in the future.** We feel **warm-grateful** for the detailed review and your time.
>
> ***
> # ***Response:***
>
> >**Question**:  Did the authors perform experiments on samples in unlabeled tasks that contain semantic mismatch data?
>
> **Answer**: Thank you for the reviewer's comments. We conducted such experiments (semantic mismatch) in our paper. In the cross-dataset generalization experiment, our training dataset is ImageNet, and our testing dataset consists of 10 other datasets (Caltech101,OxfordPets,StanfordCars,Flowers102,Food101,FGVCAircraft,SUN397,UCF101,DTD,EuroSAT). The embeddings we trained on ImageNet are relatively semantically mismatched with these other 10 datasets. Because there is a significant difference in the distribution of images between the ImageNet dataset and the remaining 10 datasets.
>
> >**Application**:  Exploration of practical applications (semantic mismatch).
>
> **Further work**:  Thank you to the reviewer for their appreciation and comprehensive comments on our work. We briefly introduce our future research (semantic mismatch) content combined with practical scenarios.
>
> * Recently, some Internet companies' e-commerce platforms frequently host live shopping events where hosts present products. However, certain individuals may alter (semantic mismatch) product packaging styles/colors during sales or position items away from the camera to conduct deceptive promotions. In our field, there exists a domain generalization experiment involving training on ImageNet and testing across four ImageNet variants. Our model shows potential applicability in scenarios involving product style tampering.
>
> * Currently, some companies utilize prompt engineering to invoke large language models. Engineers manually draft extensive prompt documentation. However, these documents may contain logical errors in the prompt scripts or inappropriate expressions (semantic mismatch) of prompt words, which can prevent the large model from completing the designated tasks. Can we substitute key components of handwritten prompt documents with inferential embeddings? Through few-shot learning with large models and reverse inference of learning embeddings, this approach might enable large models to generate more accurate results.
>
>
> >**Discussion**:  If the reviewer has further suggestions and comments, we welcome discussions with reviewer.

---

> > ### Comment · Reviewer_w5x5 · 2025-08-03
> > **Questions on some responses**
> >
> > Hi,
> >
> > Q1: Why would the authors claim Caltech101, StafordCars are semantically mismatched from the ImageNet dataset? Based on my understanding, they are all vision datasets. ImageNet should include cars and animals, vehicles, household items (Caltech101). The distribution might be different between ImageNet and Caltech101 but they are not semantically mismatched.
> >
> > For future work, I appreciate the authors list out all the possibilities.

---

> ### Author Response · Authors · 2025-08-03
> **We add the experiments of semantically mismatched**
>
> # ***Thanks to the reviewer:***
>
> > We thank the reviewers for the experimental guidance. The research direction pointed out by the reviewer is really meaningful. This will **greatly help expand the scalability of our work**. In future work, we will further study the research direction of semantic misalignment pointed out by the reviewer.
>
> ***
> ***
> # ***Response:***
>
> >**Q**:  Did the authors perform experiments on samples in unlabeled tasks that contain semantic mismatch data?
>
> **A**: We **add experiments** and design a new experimental setup. In this experiment, we trained 10 datasets (Caltech101,OxfordPets,StanfordCars,Flowers102,Food101,FGVCAircraft,SUN397,UCF101,DTD,EuroSAT) with 16-shot and 5 epochs, and then tested each dataset directly on the remaining 9 datasets. We did not use the ImageNet dataset because it has too many categories and would include other datasets.
>
> This setting is to simulate the situation of semantic mismatch. For example, after the model is trained on the Caltech101 dataset, the current embeddings are fitted to the Caltech101 dataset. Then when we use this model to test the Food101 dataset, the semantics of the text branch embeddings and the visual branch test dataset are not aligned.
>
>
> * Cross-dataset experiment by full training 10 datasets
>
> |               | [Test] |Caltech | Food| DTD | UCF | Flowers | Pets |FGVC  |  Cars| SUN | EuroSAT |
> |:----------|:------:|:------:|:-------:|:------:|:------:|:-------:|:------:|:------:|:------:|:------:|:------:|
> | **[Training]** |     |     ||||      |     |      |       |    ||
> | Caltech(94.7) | |         | 86.0|45.6 | 65.2|  67.2     |  88.3    |   23.8    |   65.5     |  65.7   | 44.6|
> | Food(87.1)  | |92.7|  | 43.3| 64.9| 67.9 | 88.3|  23.5|  65.5| 64.8| 47.5|
> | DTD(58.3)|  |93.1 |  85.6|  |  64.8|  69.8| 87.5| 22.4| 64.9| 62.0| 43.7|
> | UCF(76.5)| |91.0| 85.4| 43.7|  | 65.9| 83.1|20.9|63.7|62.8|42.8|
> | Flowers(87.5)| |89.5| 80.5| 39.1| 58.2| | 82.9|19.9| 44.6| 52.6| 51.9|
> |  Pets(93.1)  | |93.1| 86.2 | 42.1| 63.5| 66.4| | 21.6| 44.6| 52.6| 48.7|
> |  FGVC(31.0)  | |81.3|84.1|40.7|63.3|62.4|83.2||44.6|58.2|45.0|
> |  Cars(69.2)  |   | 90.3 |85.2| 40.7| 62.8| 59.4| 85| 22.3| | 61.0|47.7|
> |  SUN(70.0)  | |  91.9| 85.7 | 38.4| 63.9| 63.7| 83.2| 21.7| 62.4| |49.1|
> | EuroSAT(76.7) | | 84.3| 68.7| 31.2| 54.2| 32.0| 43.6| 17.5|39.1|53.6||
>
> This type of full training cross-dataset generalization experiment has not yet been conducted in  works [1-16] in this field. According to the reviewers' guidance and professional suggestions, we found that this full training cross-dataset experiment (with semantic mismatch between text and visual branches) is very meaningful and can further analyze and reflect generalization scenarios. We will revise this part of the experiment in the **main text to thank the reviewer for the valuable time**.
>
> ***
>
> # ***Discussion:***
>
> >If the reviewer has further suggestions and comments, we welcome discussions with reviewer.
>
>
>
> ***
> [1]Consistent prompt learning for vision-language models
>
> [2]Consistency-guided prompt learning for vision-language models
>
> [3]Self-regulating prompts: Foundational model adaptation without forgetting
>
> [4]Learning to prompt for vision-language models
>
> [5]Visual prompt tuning
>
> [6]Plot: Prompt learning with optimal transport for vision-language models
>
> [7]Domain prompt learning with quaternion networks
>
> [8]Tuning multi-mode token-level prompt alignment across modalities
>
> [9]Conditional Prompt Learning for Vision-Language Models
>
> [10]Visual language prompt tuning with knowledge-guided context optimization
>
> [11]Textual based class-aware prompt tuning for visual-language model
>
> [12]Domain prompt learning with quaternion networks
>
> [13]Few-shot adversarial prompt learning on vision-language models
>
> [14]Tree of attributes prompt learning for vision-language models
>
> [15]Dept: Decoupled prompt tuning
>
> [16]Prompt distribution learning

---

> > ### Author Response · Authors · 2025-08-03
> > **Welcome for more discussions**
> >
> > Dear Reviewer,
> >
> > We sincerely appreciate your dedicated time and effort in reviewing our work.
> >
> > If you have any additional questions or issues that require further clarification, please do not hesitate to let us know. We would be more than happy to address them promptly.
> >
> > Thank you once again for your invaluable support and contributions to improving our work. We greatly appreciate your feedback.
> >
> > Best regards,
> >
> > Authors of #27145

---

> ### Author Response · Authors · 2025-08-06
> **Would you mind raising the score**
>
> Dear reviewer,
>
> Your constructive comments have greatly helped us improve our paper. Do you have any other concerns right now? If you have no further questions/concerns, would you mind raising the score? Your evaluation of our work is invaluable and we greatly appreciate your time.
>
> Best regards and thanks,
>
> Authors

---

> ### Author Response · Authors · 2025-08-07
> **Window for discussion is closing**
>
> Dear Reviewer,
>
> We sincerely appreciate your dedicated time and effort in reviewing our work.
>
> If you have any additional questions or issues that require further clarification, please do not hesitate to let us know. We would be more than happy to address them promptly.
>
> Thank you once again for your invaluable support and contributions to improving our work. We greatly appreciate your feedback.
>
> Best regards,
>
> Authors of #27145

---

> ### Author Response · Authors · 2025-08-08
> **Could you please consider improving score?**
>
> Dear reviewer,
>
> Your constructive comments have greatly helped us improve our paper. Do you have any other concerns right now? If you have no further questions/concerns, would you mind raising the score? Your evaluation of our work is invaluable.
>
> Best regards and thanks,
>
> Authors

---

> > ### Comment · Reviewer_w5x5 · 2025-08-08
> > **Thanks for your rebuttal.**
> >
> > Hi, I appreciate the authors list out additional experiments. While I appreciate the scalability of experiments, the main innovation I understand is changing the training loss as Equation (6) denotes without theoretical justification. $\gamma_{1}$ and $\gamma_{2}$ are hyperparameters that require tuning. Therefore, I would incline to borderline accept.

---

### Official Review · Reviewer_osmA · 2025-06-09

**Clarity:** 3
**Significance:** 3
**Originality:** 3
**Rating:** 4
**Confidence:** 2

**Summary:**

This paper addresses the suboptimal parameter configuration issue in unlabeled tasks. The authors propose a plug-in method named Levelling Paradigm (LePa) to alleviate the problem of uncoordinated learning in such scenarios. The proposed method is compatible with vision-language prompting frameworks and dynamically constrains the objective functions of two-modal alignment and visual alignment. The method is evaluated on 11 real-world unlabeled tasks, and demonstrates superiority over existing approaches in multiple aspects.

**Questions:**

1. Why might the two objectives,  $L_{CE}$  and $L_{Visual}$ exhibit uncoordinated learning? Is there an intuitive explanation for this phenomenon?

2. Since the method relies on manually crafted prompts, what would happen if fully automatic prompts (e.g., generated by LLMs) were used instead? Would the performance improve or degrade?

**Ethical Concerns:**

["NO or VERY MINOR ethics concerns only"]

**Final Justification:**

The author's reply addresses my main concern

**Limitations:**

yes

**Quality:**

3

**Strengths And Weaknesses:**

Strengths:

1. The paper is clearly written and logically organized, with no significant grammatical issues.

2. The proposed method shows strong generality and can be integrated as a plug-in into various VLP frameworks.

3. The experimental results are generally strong across most tasks, although there are a few cases where the performance does not meet expectations.

Weeknesses:

1. The method shows limited performance on the source dataset (labeled ImageNet) used for training.

2. The approach relies on several hyperparameters, which may affect reproducibility and tuning difficulty.

3. The proposed method heavily depends on manually crafted prompts. It remains unclear how well it would perform under fully automatic prompt generation.

---

> ### Author Rebuttal · Authors · 2025-07-29
>
> # ***Thanks to the reviewer:***
> > The comments provided by the reviewers are very useful for our work, **we thank the reviewer for the valuable time and consideration of our manuscript, we express our heartfelt gratitude.**
>
> ***
> # ***Response:***
>
> >**Q1**: Training on the source dataset (labeled ImageNet).
>
> **A1**: Thank you very much to the reviewer for important comments. This experiment for training ImageNet is not our focusing experiment in the future, our focusing experiment is the generalization experiment (performance on 10 datasets and 4 variant datasets). The theme of our paper is to address generalization tasks.
>
>
>
> >**Q2**: What would happen if fully automatic prompts (e.g., generated by LLMs) were used instead? Would the performance improve or degrade?
>
> **A2**: The experiments proposed by the reviewer are very **meaningful** and have greatly helped our work. We will include this part of the experiments in the main paper and express our **gratitude** to reviewer. If the reviewer has further suggestions on ablation experiments, we welcome discussions with reviewer.
>
> * According to the reviewer's comments, we **add experiments**. We use LLM to generate some prompt word templates, with a quantity of 60, which is consistent with the quantity provided by CLIP official. After experimentation, we found that the effect of using LLM did not exceed that provided by the official CLIP library, although there was no significant difference between the LLM and CLIP library. We analyzed that it may be due to the incompatibility between the template generated by LLM and the pre-trained CLIP. Our 60 fixed handwritten prompt word templates are more deterministic because they come from the official CLIP repository. Compared to calling an external LLM to generate prompt words, our method is more reusable and concise.
>
> * Domain generalization
> | Method | ImageNet (Training) | -V2| -S | -A | -R | Average |
> |:------ |:------:|:-------:|:------:|:------:|:-------:|:------:|
> | 1) PromptSRC | 71.27 | 64.35 |49.55 | 50.90 | 77.80 | 60.65 |
> | 2) PromptSRC + Ours(LLM)  | 71.44 | 65.11 |50.03 | 51.83| 78.30 |  61.31 |
>
>
>
> * Cross-dataset generalization
> | Method| ImageNet (Training) | Average |
> |:------|:------:|:------:|
> | 1) PromptSRC  | 71.27 |  65.81 |
> | 2) PromptSRC + Ours(LLM)  | 71.44 |  66.43 |
>
>
>
>
>
> >**Q3**:  Why might the two objectives, Loss-CE and Loss-visual exhibit uncoordinated learning? Is there an intuitive explanation for this phenomenon?
>
> **A3**: Thank you for the useful questions raised by the reviewer.
>
> * The relationship between generalization and multi-loss uncoordinated learning has been experimentally and empirically demonstrated in literature [3, 6, 7]. The theoretical basis is that the purpose of CE loss is to fine tune and enhance few-shot ability [9], while the purpose of visual loss is to avoid forgetting pre-trained features and enhance generalization ability [3,6,7]. There is a compromise and balance between these two losses, a natural conflict where one goes up and the other goes down [8]. To prevent visual loss convergence instability and affect CE loss, we propose an HCE that normalizes HCE and visual loss, using this method to maintain the stability of visual loss. HCE is composed of pre-trained manual text features and has generalization ability [1,2,3,6,7].
>
> * Phenomenon: In the convergence process of visual loss and CE loss, there will be fluctuations in visual loss, specifically manifested as: At the beginning of training, visual loss is sometimes very small, and the CE loss is relatively large at this time. At the end of training, the CE loss converges quickly and relatively small, and the visual loss is sometimes relatively large and converges slowly. The convergence of CE loss is stable, and the convergence of visual loss is unstable.  Now, through our method, both visual loss and CE loss have steadily decreased. We have alleviated this phenomenon of uncoordinated learning.
>
> ***
>
> [1]Visual language prompt tuning with knowledge-guided context optimization
>
> [2]Textual based class-aware prompt tuning for visual-language model
>
> [3]Consistent prompt learning for vision-language models
>
> [4]Tree of attributes prompt learning for vision-language models
>
> [5]Tuning multi-mode token-level prompt alignment across modalities
>
> [6]Consistency-guided prompt learning for vision-language models
>
> [7]Self-regulating prompts: Foundational model adaptation without forgetting
>
> [8]Dept: Decoupled prompt tuning
>
> [9]Learning to prompt for vision-language models

---

> > ### Comment · Reviewer_osmA · 2025-08-06
> >
> > Thanks for the author's rebuttal. After reading it, I still think the performance and experimental design of this approach still has some flaws. Tables 2 and 3 in the paper, as well as the tables provided by Rebuttal, all show little improvement compared to the previous SOTA method (less than 1% accuracy).
> > Therefore, I decided to maintain my score.

---

> ### Author Response · Authors · 2025-08-04
> **Window for discussion and revision is closing**
>
> Dear Reviewer,
>
> Thanks a lot for your time in reviewing and insightful comments, according to which we have carefully revised the paper to answer the questions. We sincerely understand you’re busy. But since the discussion due is approaching, would you mind checking the response to confirm where you have any further questions?
>
> We are looking forward to your reply and happy to answer your further questions.
>
> Best regards
>
> Authors

---

> ### Comment · Area_Chair_95bD · 2025-08-05
>
> Dear Reviewer osmA,
>
> Please respond to authors’ rebuttals beyond Mandatory Acknowledgement and participate in discussions. Thanks.
>
> Best regards,
> AC

---

> ### Author Response · Authors · 2025-08-06
> **Response**
>
> # ***We thank the reviewer:***
> > We thank the reviewer for the valuable time and consideration of our manuscript, we express our heartfelt gratitude.
>
> ***
> ***
> # ***Response:***
>
> >**Q1**: Tables 2 and 3 in the paper, show little improvement compared to the previous SOTA method.
>
> **A1**: We thank the reviewers for the experimental guidance.
> * In our paper, the mean in Table 1 is 61.68%, while TAP + DePT is 60.61, our performance gain is **over 1%**. In Table 2, our mean is 67.18, while TAP + DePT is 66.20, our performance gain is **0.98%, very close to 1%**.
>
> * **It is worth noting that it is difficult to improve the performance of this settings of generalization experiment [1-16]**. In the domain generalization experiment shown in Table 1, PromptSRC (ICCV2023), a work by others, only achieved a 0.4% improvement over MaPLe (CVPR2023). In the cross-dataset generalization experiment shown in Table 2, PromptSRC (ICCV2023), a work by others, performed worse than MaPLe (CVPR2023).
>
> * TAP (ICLR2025) performs worse than MaPLe (CVPR023) in both cross-dataset generalization and domain generalization experiments. Furthermore, DePT's performance gain over Maple, PromptSRC, and TAP is less than 0.5% in both generalization experiments.
>
>
> >**Q2**: The tables provided by Rebuttal (LLM, other method), show little improvement compared to the previous SOTA method.
>
> **A2**: We are very grateful to the reviewer for proposing the LLM method. However, the LLM generation method is not suitable for our design. The core of our method is to fully explore the manual templates provided by CLIP. However, the LLM generation is not compatible with our core idea.
>
> ***
> ***
>
> # ***Discussion***:
>
> We hope that the reviewers can provide a **second evaluation of the performance improvement** of our work and we are very grateful for the reviewers' time.
>
>
>
> ***
>
> [1]Consistent prompt learning for vision-language models
>
> [2]Consistency-guided prompt learning for vision-language models
>
> [3]Self-regulating prompts: Foundational model adaptation without forgetting
>
> [4]Plot: Prompt learning with optimal transport for vision-language models
>
> [5]Domain prompt learning with quaternion networks
>
> [6]Tuning multi-mode token-level prompt alignment across modalities
>
> [7]Conditional Prompt Learning for Vision-Language Models
>
> [8]Visual language prompt tuning with knowledge-guided context optimization
>
> [9]Textual based class-aware prompt tuning for visual-language model
>
> [10]Domain prompt learning with quaternion networks
>
> [11]Few-shot adversarial prompt learning on vision-language models
>
> [12]Tree of attributes prompt learning for vision-language models
>
> [13]Dept: Decoupled prompt tuning
>
> [14]Prompt distribution learning

---

> ### Author Response · Authors · 2025-08-07
> **Window for discussion and revision is closing**
>
> Dear Reviewer,
>
> Thanks a lot for your time in reviewing and insightful comments, according to which we have carefully revised the paper to answer the questions. We sincerely understand you’re busy. But since the discussion due is approaching, would you mind checking the response to confirm where you have any further questions?
>
> We are looking forward to your reply and happy to answer your further questions.
>
> Best regards
>
> Authors

---

> > ### Comment · Reviewer_osmA · 2025-08-07
> >
> > Thank you for the author's prompt response. Despite what the author claims, I believe that a limited improvement (even the claimed 1% or so) is insufficient to demonstrate its innovation and effectiveness; it's simply a different approach to the same solution. While the performance is comparable, I'm curious about how the efficiency of the author's approach compares to Sota's. Achieving the same performance with less time or resources would, in my opinion, be a greater contribution.

---

> > > ### Author Response · Authors · 2025-08-08
> > > **Efficiency experiment**
> > >
> > > # ***We thank the reviewer:***
> > > > We thank the reviewer for the valuable time and consideration of our manuscript, we express our heartfelt gratitude.
> > >
> > > ***
> > > ***
> > > # ***Response:***
> > >
> > > >**Q1**: Achieving the same performance with less time or resources would, in my opinion, be a greater contribution.
> > >
> > > **A1**: In Table 4(a) of the paper in PromptSRC, our performance is better than DePT. Now, we have repeated the experiment and found that when our **performance** is **consistent with DePT**, our training time is 13.50 minutes, which is **less** than DePT's 13.88 minutes, and we have **not increased the number of parameters**, DePT needs to add parameters.
> > >
> > > ***
> > > # ***Dicussion:***
> > >
> > > Thanks a lot for your time in reviewing and insightful comments, according to which we have carefully revised the paper to answer the questions. We sincerely understand you’re busy. But since the discussion due is approaching, would you mind checking the response to confirm where you have any further questions?
> > >
> > > Do you have any other concerns right now? If you have no further questions/concerns, would you mind raising the score? Your evaluation of our work is invaluable and we greatly appreciate your time.
> > >
> > >
> > > Best regards
> > >
> > > Authors
> > >
> > >
> > >
> > >
> > >
> > >
> > >
> > >
> > >
> > >
> > >
> > >
> > >
> > >
> > > ***

---

> ### Author Response · Authors · 2025-08-08
> **Efficiency**
>
> # ***We thank the reviewer:***
> > We thank the reviewer for the valuable time and consideration of our manuscript, we express our heartfelt gratitude.
>
> ***
> ***
> # ***Response:***
>
> >**Q1**: Achieving the same performance with less time or resources would, in my opinion, be a greater contribution.
>
> **A1**:
>
> * In the paper, observing Table 4, our method **did not add any learnable parameters**, however, DePT requires some parameters depending on the size of the training dataset.
>
> * In addition, our method is based on the integration of other people's **work [1-7]** and does **not increase** the learning **parameters**, however, DePT is increased.
>
> * In Table 4 (a), we only trained **one minute more** than the DePT (PromptSRC), however, we did not increase the learning parameters at all, while the increase in DePT had a certain scale. In practical industrial scenarios, models need to be deployed on the end, and the number of parameters learned is more important than training time.
>
>
>
>
> ***
>
> [1]Consistent prompt learning for vision-language models
>
> [2]Consistency-guided prompt learning for vision-language models
>
> [3]Self-regulating prompts: Foundational model adaptation without forgetting
>
> [4]Plot: Prompt learning with optimal transport for vision-language models
>
> [5]Tuning multi-mode token-level prompt alignment across modalities
>
> [6]Few-shot adversarial prompt learning on vision-language models
>
> [7]Tree of attributes prompt learning for vision-language models

---

> > ### Comment · Reviewer_osmA · 2025-08-09
> >
> > Thanks for the author's reply. It seems addresses my main concern, so I'm willing to raise my score to weak accept.

---

### Official Review · Reviewer_uMgK · 2025-06-21

**Clarity:** 2
**Significance:** 2
**Originality:** 3
**Rating:** 3
**Confidence:** 3

**Summary:**

This paper proposes a plug-in module, Levelling Paradigm (LePa), to enhance the generalization of directly-integrated Vision-Language Prompting (VLP) frameworks, particularly in domain and cross-dataset generalization under unlabeled settings. LePa addresses the issue of uncoordinated optimization between vision-language alignment and visual feature preservation by introducing a dynamic constraint objective. The method is compatible with multiple VLP baselines and shows consistent gains across 11 datasets.

**Questions:**

Please refer to the weaknesses outlined above.

**Ethical Concerns:**

["NO or VERY MINOR ethics concerns only"]

**Final Justification:**

I sincerely appreciate the authors' rebuttal. However, it cannot sufficiently address my concerns in the following aspects:

(1) This work overlooks an important baseline termed Tuning-LePa, which directly minimizes the divergence between  $L_{CE}$ and $L_{visual}$ in line with the authors' motivation. Moreover, the authors fail to clearly explain why the proposed LePa outperforms Tuning-LePa.

(2) The convergence behavior of all losses is not described, preventing verification of the motivation behind the proposed LePa.

Given those concerns, I am inclined to **reject** this work, but I hope it can be further improved.

**Limitations:**

The limitations are included in the main text.

**Quality:**

3

**Strengths And Weaknesses:**

Strengths

1.	The proposed LePa is designed as a plug-in module, making it compatible with a variety of existing VLP methods, which improves its applicability and practical value.

2.	The method is evaluated across 11 diverse datasets covering multiple generalization scenarios (base-to-novel, domain generalization, cross-dataset), demonstrating consistent performance improvements.

3.	The writing of this paper is good.


Weaknesses

1.	Why does LePa not consider $L_{textual}$? Is there any experimental evidence supporting the claim that uncoordinated learning occurs between $L_{CE}$ and $L_{visual}$? A clear justification is crucial to support the motivation of this work.

2.	The authors claim that uncoordinated learning occurs between $L_{CE}$ and $L_{visual}$. However, it is unclear why LePa does not directly minimize the difference between those two losses. Also, this paper lacks experimental analysis or ablation studies to verify the effectiveness of this direct minimization approach.

3.	To alleviate the uncoordinated learning between two losses, a vanilla approach is to assign different hyper-parameters to control their contributions, or, more advanced methods could treat those hyper-parameters as learnable parameters. However, this paper lacks a discussion or experimental evaluation regarding such strategies, which are crucial for multi-losses optimization.

4.	Why does the proposed method ignore $L_{textual}$ in Eq. (6)? If so, it would be better to remove  “Textual” from Figure 2.

5.	To alleviate the uncoordinated learning between $L_{CE}$ and $L_{visual}$, this work introduces a new loss. However, there remains a risk of uncoordinated learning between the new loss and the original losses.

6.	Since DePT is also a plug-in method, it would be better to include a discussion in the main text comparing DePT and LePa. For example, what are the disadvantages of DePT, and how does LePa address or avoid them?

7.	In Section 4.3, what does the term embedding length refer to?

8.	In the Related Work section, it would be beneficial to provide more details about the VLP methods used in the comparison experiments, i.e., MaPLe, PromptSRC, FAP, and TAP.

Small issues:

1.	In line 87, “base-to novel” should be “base-to-novel”

2.	In line 89, “several representative VLP works incorporates” should be “several representative VLP works incorporate”

3.	In line 90, “task-specific feature” should be “task-specific features”

---

> ### Author Rebuttal · Authors · 2025-07-29
>
> # ***Thanks to the reviewer:***
>
> > We sincerely thank the reviewers' **generous time and very detailed comments**.  We understand that our current manuscript has shortcomings. We highly **respect the time and effort of reviewers. We will revise the manuscript carefully** according to the reviewers' comments.
>
> ***
> # ***Response:***
>
> >**Q1**:  Is there any experimental evidence supporting the claim that uncoordinated learning occurs between Loss-CE and Loss-visual ?
>
> **A1**:  In the convergence process of visual loss and CE loss, there will be fluctuations in visual loss, specifically manifested as: At the beginning of training, visual loss is sometimes very small, and the CE loss is relatively large at this time. At the end of training, the CE loss converges quickly and relatively small, and the visual loss is sometimes large and converges slowly. The convergence of CE loss is stable, and the convergence of visual loss is unstable [1,2,3].  The architecture of [1,2,3] methods inherently includes CE loss and visual loss.
>
> >**Q2**:  It is unclear why LePa does not directly minimize the difference between those two losses (CE loss and visual loss).
>
> **A2**:  Because through our observation, the decrease in CE loss is stable, however, the decrease in visual loss is not stable, and visual loss occasionally increases at the end of training [1,2,3]. Therefore, we do not intend to directly limit CE and visual loss. Restricting both directly would disrupt the convergence process of CE. We propose an HCE, which is the cross-entropy loss of features generated by manually prompted words. According to the paper [1,3,10,11], the features generated by the pre-trained model of manually prompted words have strong generalization ability. Therefore, we limit visual loss and manual HCE. Through this approach, visual loss is made more robust, and the instability of visual loss convergence is alleviated.
>
> >**Q3**: This paper lacks ablation studies to verify the effectiveness of this direct minimization approach.
>
> **A3**: The experiments proposed by the reviewer are very meaningful and have greatly **helped** our work. We will include this part of the experiments in the main paper and express our gratitude to reviewer. If the reviewer has further suggestions on ablation experiments, we welcome discussions with reviewer.
>
> * We **add experiment** to analyze the direct minimization approach. Our experiment is based on the classic VLP work PromptSRC [3], which has four native loss functions: textual loss, visual loss, CE loss, and KL loss. We define the constraint function between CE loss and visual loss as Tuning-LePa, without HCE and hand-crafted prompts. Our paper models the relationship between HCE loss and visual loss using the LePa constraint function. We observed the experiment and found that the direct regularization method (row-2) resulted in a decrease in the performance of training ImageNet, with minimal changes in generalization performance compared to row-1. This may be due to the fact that the direct minimization approach does not allow CE to converge well, and this approach has little effect on visual loss optimization.
>
> * Domain generalization
> | Method | ImageNet (Training)  | Average |
> |:------ |:------:|:-------:|
> | PromptSRC | 71.27  | 60.65 |
> | PromptSRC + Tuning-LePa| 71.05  | 60.80 |
>
>
> * Cross-dataset generalization
> | Method| ImageNet (Training) | Average |
> |:------|:------:|:------:|
> | PromptSRC  | 71.27 |  65.81 |
> | PromptSRC + Tuning-LePa | 71.05 |  65.85 |
>
>
> >**Q4**:   This paper lacks a discussion or experimental evaluation regarding such strategies (different hyper-parameters to control their contributions).
>
> **A4**:  The experiments proposed by the reviewer are very meaningful. We will include this part of the experiments in the **main paper and express our gratitude to reviewer**. If the reviewer has further suggestions on ablation experiments, we welcome discussions with reviewer.
>
> * We **add experiments** for the visual loss and CE loss to explore the multi-losses optimization. We add a hyperparameter $\gamma_{3}$ (0.3,3) to control the original CE loss. In addition, we use the hyperparameter $\gamma_{2}$ (10) to control visual loss. From the experiments on generalization metrics, we found that strengthening CE loss will enhance the performance of training ImageNet and reduce its generalization performance. Weakening CE loss will reduce the performance of ImageNet and enhance its generalization performance. However, the variation in generalization performance is limited, and the performance of training ImageNet shows significant changes.
>
> * Domain generalization
> | Method | ImageNet (Training)  | Average |
> |:------ |:------:|:-------:|
> | 1) PromptSRC | 71.27  | 60.65 |
> | 2) PromptSRC (10 * visual loss + 0.3 * CE) | 70.11 | 60.82 |
> | 3) PromptSRC (10 * visual loss + 3 * CE) | 71.35  | 60.46 |
>
>
> * Cross-dataset generalization
> | Method| ImageNet (Training) | Average |
> |:------|:------:|:------:|
> | 1) PromptSRC  | 71.27 |  65.81 |
> | 2) PromptSRC (10 * visual loss + 0.3 * CE  | 70.11 |  66.00 |
> | 3) PromptSRC (10 * visual loss + 3 * CE) | 71.35 |  65.01 |
>
>
> >**Q5**:   Why does the proposed method ignore Loss-textual? If so, it would be better to remove “Textual” from Figure 2.
>
> **A5**:  We employ a plug-in technology. Our plug-in technology does not need textual loss. The lower box in Figure 2 describes PromptSRC, which is an example baseline. PromptSRC needs textual loss. However, some VLP methods do not have textual loss functions. We apologize for our vague architecture and unclear explanation. We promise to make careful revisions.
>
> >**Q6**:   It would be better to include a discussion in the main text comparing DePT and LePa.
>
> **A6**:  According to the reviewers' opinions, we will discuss it in the main paper to give readers more introduction.
>
> * The DePT method increases the number of learning parameters, which are related to the parameters of the base class. However, our plug-in does not add a new number of learnable parameters. Our component only adds the type of loss.
>
> * The DePT method does not perform domain generalization experiments in the main paper of DePT. Our domain generalization experiments are in the main paper. Our method has carried out three classical generalization experiments, which are more than the generalization experiments of DePT in the paper, and our indicators are more comprehensive. In addition, in our domain experiment, the performance is about 1% higher than that of DePT.
>
> >**Q7**:   In Section 4.3, what does the term embedding length refer to?
>
> **A7**:  VLP architecture has visual embeddings and textual embeddings. These embeddings need to be learned. Among them, the length of embeddings, that is, the length of token, is a hyper-parameter setting. For example, if the hyper-parameter of the embeddings length of this text branch is 4, then the parameter quantity of this part is "4 * 512 = 2048". It is worth noting that the length of the embeddings cannot be too long, because in this case, the model will over fit the downstream data set [9]. There is a threshold value for the length of the embeddings, which needs to be adjusted to constrain the generalization tasks and non generalization tasks. This hyper-parameter also exists in these works [1-11].
>
> >**Q8**:   It would be beneficial to provide more details about the VLP methods used in the comparison experiments.
>
> **A8**:  According to the opinions of the reviewers, we will add text introduction to the relevant work description,  so as to make the readers understand our work clearly.
>
> * MaPLe uses depth embeddings technology in both visual encoder and text encoder, that is, there are two super-parameters of embeddings: depth and length. MaPLe transfers text embeddings to the visual branch as a visual embeddings through a transformation function that can be learned by transformation. Please note that MaPLe's visual embeddings are not initialized, but transferred from other places. The PromptSRC  aims to add some regularization loss functions instead of learning parameters in the VLP structure. PromptSRC has no connected function between two encoders. In detail, PromptSRC adopts four kinds of losses: 1) the loss of text fine-tuning and non fine-tuning, 2) the loss of visual fine-tuning and non fine-tuning, 3) the loss of conventional fine-tuning CE, and 4) the KL loss of fine-tuning CE and non fine-tuning CE.
> FAP uses the method of resisting attacks to increase the image disturbance. The cosine value generated by natural image and disturbed image is limited by KL divergence. There are some parameters to be learned, which is the same as the middle layer function of MaPLe. TAP first instructs LLMs to generate a tree of attributes with a “concept - attribute - description” structure for each category, and then learn the hierarchy with vision and text prompt tokens. TAP essentially distills structured knowledge graphs associated with class names from LLMs.
>
>
> ***
>
> [1]Consistent prompt learning for vision-language models
>
> [2]Consistency-guided prompt learning for vision-language models
>
> [3]Self-regulating prompts: Foundational model adaptation without forgetting
>
> [4]Learning to prompt for vision-language models
>
> [5]Visual prompt tuning
>
> [6]Plot: Prompt learning with optimal transport for vision-language models
>
> [7]Domain prompt learning with quaternion networks
>
> [8]Tuning multi-mode token-level prompt alignment across modalities
>
> [9]Conditional Prompt Learning for Vision-Language Models
>
> [10]Visual language prompt tuning with knowledge-guided context optimization
>
> [11]Textual based class-aware prompt tuning for visual-language model

---

> > ### Comment · Reviewer_uMgK · 2025-08-04
> >
> > We appreciate your detailed response. My major issues have been addressed. There are remaining concerns that need to be addressed.
> >
> > 1. Regarding A1, could you please specify which dataset was used to observe this phenomenon? Is this phenomenon commonly observed across most datasets? This clarification is quite important as it forms a core motivation for the work. Additionally, including the relevant experimental results in both the Introduction and Experiment sections would strengthen the manuscript. However, **the lack of these results in the current version of the manuscript constitutes a significant omission**.
> >
> > 2. Regarding A3, was any hyper-parameter selection conducted, or **was the hyper-parameter simply set to 1**?
> >
> > 3. Regarding A4, only limited values for $\gamma_3$  (0.3, 3) and $\gamma_2$  (10) were used, which may not be sufficient to fully verify the effects of the different losses. **A broader range of values** would provide a more comprehensive evaluation.
> >
> > 4. The authors did not address my concern regarding point 5. **This is an important issue, as a new loss function is introduced**. It is essential to clarify how the proposed loss avoids the risk of uncoordinated learning.
> >
> > Given that the above concerns, I am inclined to keep my original score.

---

> ### Author Response · Authors · 2025-08-04
> **Window for discussion is closing**
>
> Dear Reviewer,
>
> Thanks a lot for your time in reviewing and insightful comments, according to which we have carefully revised the paper to answer the questions. We sincerely understand you’re busy. But since the discussion due is approaching, would you mind checking the response to confirm where you have any further questions?
>
> We are looking forward to your reply and happy to answer your further questions.
>
> Best regards
>
> Authors

---

> ### Author Response · Authors · 2025-08-05
> **Our responses**
>
> # ***We thank the reviewer:***
>
>
> > Thanks for your great efforts in reviewing our paper. Your constructive comments have greatly helped us improve our paper. We **promise to revise carefully**.
>
> ***
> ***
>
> # ***Response:***
>
> >**Q1**:  Could you please specify which dataset was used to observe this phenomenon? Is this phenomenon commonly observed across most datasets? These results should be presented in the main text.
>
> **A1**: This situation primarily occurs on the ImageNet dataset, and in both cross-dataset generalization and domain generalization experiments. In both generalization experiments, we trained on the ImageNet dataset, which has 1000 categories, randomly selecting 16 images from each category. The trained model was then tested directly on the remaining 10 datasets and four variant datasets, rather than training on the remaining 10 datasets and four variant datasets. In these two generalization experimental settings, our work and others only train on ImageNet, and the convergence instability of visual loss has nothing to do with the type of dataset, but is caused by the combination of (CE fine-tuning loss + visual generalization loss)[1][2][3]. The current rebuttal does not support uploading images and PDF. We promise to add a line graph of the visual loss of the ImageNet training process with 20 training epochs in the revised version, and then put this part of the experiment in the introduction.
>
>
>
> >**Q2**:   Was any hyper-parameter selection conducted in no LePa?
>
> **A2**: This part of the experiment is based on PromptSRC. In PromptSRC, they set the weight of textual loss to 25 and the weight of visual loss to 10. In addition, they also set the KL divergence of pre-training loss and fine-tuning loss, which has a weight of 1. And, the weight of CE loss is 1.  The weight of the Tuning-LePa module we improved is set to 1.
>
>
>
> >**Q3**:   Regarding A4, only limited values for
>  $\gamma_1$ (0.3, 3) and $\gamma_3$ (10) were used, which may not be sufficient to fully verify the effects of the different losses.
>
>
> **A3**: We now add ablation settings to produce a more extensive evaluation.
>
> * Domain generalization
> | Method | ImageNet (Training)  | Average of 4 datasets |
> |:------ |:------:|:-------:|
> | 1) PromptSRC (10 * visual loss + 0.3 * CE) | 70.11 | 60.82 |
> | 2) PromptSRC (10 * visual loss + 1 * CE) | 71.27  | 60.65 |
> | 3) PromptSRC (10 * visual loss + 2 * CE) | 71.36  | 60.33 |
> | 4) PromptSRC (10 * visual loss + 3 * CE) | 71.35  | 60.46 |
> | 5) PromptSRC (10 * visual loss + 4 * CE) | 72.01  | 60.03 |
> | 6) PromptSRC (5 * visual loss + 4 * CE) | 72.10  | 59.85 |
> | 7) PromptSRC (3 * visual loss + 4 * CE) | 72.15  | 59.60 |
> | 8) PromptSRC (10 * visual loss + 1 * CE) + Tuning-LePa| 71.05  | 60.80 |
> | 9) PromptSRC (10 * visual loss + 2 * CE) + Tuning-LePa| 71.09  | 60.51 |
> | 10) PromptSRC (5 * visual loss + 2 * CE) + Tuning-LePa| 71.13  | 60.18 |
>
>
> * Cross-dataset generalization
> | Method| ImageNet (Training) | Average of 10 datasets |
> |:------|:------:|:------:|
> | 1) PromptSRC (10 * visual loss + 0.3 * CE）  | 70.11 |  66.00 |
> | 2) PromptSRC (10 * visual loss + 1 * CE) | 71.27 |  65.81 |
> | 3) PromptSRC (10 * visual loss + 2 * CE) | 71.36 |  65.66 |
> | 4) PromptSRC (10 * visual loss + 3 * CE) | 71.35 |  65.01 |
> | 5) PromptSRC (10 * visual loss + 4 * CE) | 72.01 |  65.32 |
> | 6) PromptSRC (5 * visual loss + 4 * CE) | 72.10  | 64.33 |
> | 7) PromptSRC (3 * visual loss + 4 * CE) | 72.15  | 64.05 |
> | 8) PromptSRC (10 * visual loss + 1 * CE) + Tuning-LePa | 71.05 |  65.85 |
> | 9) PromptSRC (10 * visual loss + 2 * CE) + Tuning-LePa | 71.09 |  65.10 |
> | 10) PromptSRC (5 * visual loss + 2 * CE) + Tuning-LePa | 71.13 |  64.35 |
>
>
>
> >**Q4**:   To alleviate the uncoordinated learning between visual loss and CE loss, this work introduces a new loss. However, there remains a risk of uncoordinated learning between the new loss and the original losses. This is an important issue, as a new loss function is introduced.
>
>
> **A4**: The newly proposed loss is a combination of the HCE loss and the visual loss. Our motivation is to focus solely on the convergence of the visual loss and the CE loss. More specifically, we focus solely on the convergence process of the visual loss because we have observed that the overall convergence rhythm of the CE loss is stable. Although we propose a new loss, LePa, we do not consider whether the LePa loss learns inconsistently with the visual loss or other losses. We only focus on whether both the CE loss and the visual loss decrease smoothly. After integrating LePa, we found that the visual loss decreased smoothly, and from the two generalization experiments, the generalization performance was improved.
>
>
> ***
>
> [1]Consistent prompt learning for vision-language models
>
> [2]Consistency-guided prompt learning for vision-language models
>
> [3]Self-regulating prompts: Foundational model adaptation without forgetting

---

> ### Author Response · Authors · 2025-08-06
> **Would you mind raising the score**
>
> Dear reviewer,
>
> Your constructive comments have greatly helped us improve our paper. Do you have any other concerns right now? If you have no further questions/concerns, would you mind raising the score? Your evaluation of our work is invaluable and we greatly appreciate your time.
>
> Best regards and thanks,
>
> Authors

---

> > ### Comment · Reviewer_uMgK · 2025-08-07
> >
> > Thank you for your thorough response. I greatly appreciate the time and effort dedicated to the discussion. However, I would like to seek further clarification on a few specific points.
> >
> > 1. Regarding A2, there is a lack of **the hyper-parameter tuning experiments for the weight of the Tuning-LePa module**. If using Tuning-LePa already leads to improved performance, then the necessity of introducing LePa becomes questionable. This point is very important to verify your claim that  **restricting both directly would disrupt the convergence process of CE**. According to the results present in the rebuttal (i.e., in A3, comparisons 2) and 8) in both tables), **I do not find a bad impact in introducing Tuning-LePa, and introducing Tuning-LePa brings a slight performance improvement**. Therefore, could you provide the experimental results for this weight from the set {0.1, 5, 10, 20, 50}?
> >
> > 2. Regarding A3, could you kindly provide the experimental results with the following settings: (10 × visual loss + 0.1 × CE), (10 × visual loss + 0.01 × CE), (10 × visual loss + 0.05 × CE) , and (10 × visual loss + 0.005 × CE)? **It appears that increasing the weight of the CE loss tends to negatively impact performance**.
> >
> > 3. Regarding A4: It would be more convincing to provide convergence curves of $L\_{\\text{CE}}$ and $L\_{\\text{visual}}$ before and after applying LePa. Additionally, including curves of $L\_{\\text{HCE}}$ and  $L\_{\\text{LePa}}$ would help analyze the underlying reasons. **This is important for empirically validating your motivation, rather than solely focusing on performance gains**.

---

> > > ### Author Response · Authors · 2025-08-07
> > > **Window for discussion is closing**
> > >
> > > Dear Reviewer,
> > >
> > > We sincerely appreciate your dedicated time and effort in reviewing our work.
> > >
> > > If you have any additional questions or issues that require further clarification, please do not hesitate to let us know. We would be more than happy to address them promptly.
> > >
> > > Thank you once again for your invaluable support and contributions to improving our work. We greatly appreciate your feedback.
> > >
> > > Best regards,
> > >
> > > Authors of #27145

---

> ### Author Response · Authors · 2025-08-07
> **Experimental Response**
>
> # ***We thank the reviewer:***
>
>
> > We would like to express our gratitude to the reviewers:
>
> * providing rich and comprehensive comments for our work,
>
> * investing a significant amount of time and effort in reviewing our work,
>
> * providing detailed practical guidances, which has greatly improved our work.
>
> ***
> ***
>
> # ***Response:***
>
> >**Q1**:  Could you provide the experimental results for this weight from the set {0.1, 5, 10, 20, 50} for Tuning-LePa?
>
> **A1**: As shown in the following table, adding Tuning-LePa will lead to a decrease in the performance of ImageNet training, which is directly related to CE loss [1,2]. Disturbing the CE loss will result in a decrease in performance on the training ImageNet [1,2]. After observing the second and eighth rows, we find that there is indeed some improvement in the generalization index, but it is very small.
>
> * Now we add the ablation experiment in **row-11,row-12,row-13,row-14,row-15** for the weight of Tuning-LePa {0.1, 5, 10, 20, 50}. Experimental observations show that as the Tuning-LePa weight increases, the performance of ImageNet training gradually decreases. However, the generalization performance fluctuates and ultimately decreases.
>
> * Domain generalization
> | Method | ImageNet (Training)  | Average of 4 datasets |
> |:------ |:------:|:-------:|
> | 1) PromptSRC (10 * visual loss + 0.3 * CE) | 70.11 | 60.82 |
> | **2) PromptSRC (10 * visual loss + 1 * CE)** | 71.27  | 60.65 |
> | ***  |  *** |  *** |
> | **8) PromptSRC (10 * visual loss + 1 * CE) + 1 * Tuning-LePa**| 71.05  | 60.80 |
> | ***  |  *** |  *** |
> | 11) PromptSRC (10 * visual loss + 1 * CE) + 0.1 * Tuning-LePa| 71.18  | 60.64 |
> | 12) PromptSRC (10 * visual loss + 1 * CE) + 5 * Tuning-LePa| 71.05  | 60.90 |
> | 13) PromptSRC (10 * visual loss + 1 * CE) + 10 * Tuning-LePa| 71.02  | 60.95 |
> | 14) PromptSRC (10 * visual loss + 1 * CE) + 20 * Tuning-LePa| 69.15  | 60.85 |
> | 15) PromptSRC (10 * visual loss + 1 * CE) + 50 * Tuning-LePa| 69.08  | 60.30 |
>
> * Cross-dataset generalization
> | Method| ImageNet (Training) | Average of 10 datasets |
> |:------|:------:|:------:|
> | 1) PromptSRC (10 * visual loss + 0.3 * CE）  | 70.11 |  66.00 |
> | **2) PromptSRC (10 * visual loss + 1 * CE)**| 71.27 |  65.81 |
> | ***| *** |  *** |
> | **8) PromptSRC (10 * visual loss + 1 * CE) + 1 * Tuning-LePa** | 71.05 |  65.85 |
> | ***  |  *** |  *** |
> | 11) PromptSRC (10 * visual loss + 1 * CE) + 0.1 * Tuning-LePa| 71.18  | 65.80 |
> | 12) PromptSRC (10 * visual loss + 1 * CE) + 5 * Tuning-LePa| 71.05  | 65.88 |
> | 13) PromptSRC (10 * visual loss + 1 * CE) + 10 * Tuning-LePa| 71.02  | 66.05 |
> | 14) PromptSRC (10 * visual loss + 1 * CE) + 20 * Tuning-LePa| 69.15  | 65.81 |
> | 15) PromptSRC (10 * visual loss + 1 * CE) + 50 * Tuning-LePa| 69.08  | 64.00 |
>
> >**Q2**:   Could you kindly provide the experimental results with the following settings: (10 × visual loss + 0.1 × CE), (10 × visual loss + 0.01 × CE), (10 × visual loss + 0.05 × CE) , and (10 × visual loss + 0.005 × CE) in  PromptSRC?
>
> **A2**: We add the experiment in **row-16,row-17,row-18,row-19** for the weight of CE {0.1, 0.01, 0.05, 0.005} in PromptSRC. Experiments show that reducing the weight of CE in the PromptSRC method leads to a gradual and significant decline in ImageNet training performance. However, generalization performance fluctuates and changes are not obvious.
>
> * Domain generalization
> | Method | ImageNet (Training)  | Average of 4 datasets |
> |:------ |:------:|:-------:|
> | 1) PromptSRC (10 * visual loss + 0.3 * CE) | 70.11 | 60.82 |
> | 2) PromptSRC (10 * visual loss + 1 * CE) | 71.27  | 60.65 |
> | ***  |  *** |  *** |
> | 16) PromptSRC (10 * visual loss + 0.1 * CE) | 70.13  | 60.83 |
> | 17) PromptSRC (10 * visual loss + 0.01 * CE) | 69.91  | 60.75 |
> | 18) PromptSRC (10 * visual loss + 0.05 * CE) | 69.85  | 60.90 |
> | 19) PromptSRC (10 * visual loss + 0.005 * CE) | 69.01  | 60.10 |
>
>
> * Cross-dataset generalization
> | Method| ImageNet (Training) | Average of 10 datasets |
> |:------|:------:|:------:|
> | 1) PromptSRC (10 * visual loss + 0.3 * CE）  | 70.11 |  66.00 |
> | 2) PromptSRC (10 * visual loss + 1 * CE) | 71.27 |  65.81 |
> | ***  |  *** |  *** |
> | 16) PromptSRC (10 * visual loss + 0.1 * CE) | 70.13 |  66.05 |
> | 17) PromptSRC (10 * visual loss + 0.01 * CE) | 69.91 |  66.00 |
> | 18) PromptSRC (10 * visual loss + 0.05 * CE) | 69.85 |  66.01 |
> | 19) PromptSRC (10 * visual loss + 0.005 * CE) | 69.01 |  65.56 |
>
>
> >**Q3**:  It would be more convincing to provide convergence curves.
>
> **A3**: The reviewer's suggestion is very reasonable. Indeed, adding the loss function change curve will make our motivation more convincing. It is not only necessary to show that the experiment is effective, but also to show whether the problem is solved from the root cause. We **promise** to add the change curves of HCE, CE, visual loss, and lepa in the revised version.
>
> ***
> ***
>
> [1]Learning to prompt for vision-language models
>
> [2]Plot: Prompt learning with optimal transport for vision-language models

---

> > ### Comment · Reviewer_uMgK · 2025-08-08
> >
> > I sincerely appreciate your detailed responses. According to the new results provided by the authors, I still have major concerns.
> >
> > 1. I find that introducing Tuning-LePa alone can improve the generalization performance on target tasks to some extent. For example, on the Domain Generalization task, under the setting of PromptSRC (10 × visual loss + 1 × CE) + **10 × Tuning-LePa**, the average performance across 4 datasets is **60.95**, which outperforms PromptSRC (**60.65**) and PromptSRC + DePT (**60.93**) as reported in Table 1. Similarly, on the cross-dataset generalization task, under the setting of PromptSRC (10 × visual loss + 1 × CE) + **10 × Tuning-LePa**, the average performance across 10 datasets is **66.05**, which surpasses PromptSRC (**65.81**) and PromptSRC + DePT (**66.02**) as reported in Table 2. **Although the performance on the source task slightly declines, these results verify the generalization ability of Tuning-LePa**. Accordingly, **the claim that directly minimizing Tuning-LePa would disrupt the convergence process of CE may not be correct**. While the source task performance drops slightly, the target task performance does not. Moreover, **to assess the impact on convergence, the authors should plot the convergence curves rather than only focusing on the test performance of the source task**. **Since the source task performance decreases only marginally, I do not believe that the CE loss convergence curve would exhibit significant fluctuations**.
> >
> > 2. In view of the above analysis, **Tuning-LePa should be considered an important baseline method**, as it represents a straightforward implementation that aligns with the authors’ motivation. However, **the manuscript overlooks this important baseline and fails to clearly explain why the proposed LaPa outperforms Tuning-LePa**.
> >
> > 3. In A3, the authors did not describe the convergence behavior of all losses. Therefore, **I am unable to determine whether all losses have converged properly to substantiate the authors’ motivation**.
> >
> > Given that substantial revisions are required, I maintain my original score. Thank you.

---

### Official Review · Reviewer_Eibv · 2025-07-12

**Clarity:** 2
**Significance:** 2
**Originality:** 2
**Rating:** 4
**Confidence:** 3

**Summary:**

This paper proposes the Leveling Paradigm (LePa) loss, an additional regularization term that can be plugged into the prompt-tuning of CLIP models for image classification. Prompt-tuning with CLIP has been shown to result in overfitting to the target-task making the learned model worse at image-classification than the base CLIP model; various methods have attempted to mitigate this by adding additional regularization terms to keep the representations generated from the learnable prompts close to the representations generated without (e.g. keeping the learned representations close to the pre-trained representations). However, balancing these two objectives can be difficult, leading to what the authors term unbalanced learning. LePa is designed to address unbalanced learning by adding an additional constraint to ensure that the visual regularization term and the task-specific cross-entropy loss decrease at similar rates.

More specifically, LePa operates as a plugin to the traditional prompt-tuning CLIP paradigm; it computes an additional cross-entropy loss, termed the "Robust Cross-Entropy Loss", which is constructed using the textual features of the class prompts (without the prompt-tuning parameters) and the image representation from the image encoder with the learnable image prompt. The HCE is computed as the cross-entropy between the ground truth label and the cosine-similarity between the image representation and the k-worst class prompts (assuming a set of class prompts are being used for diversity as in PromptSRC). The LePa loss is the constructed by taking the L1 norm of the difference of the HCE loss and the visual regularization loss; in other words, the LePa loss is designed to ensure that the visual regularization term and the HCE loss decrease at similar rates.

The authors validate LePA as a plugin for various CLIP-tuning techniques, including PromptSRC, FAP, TAP, and MaPLe. They show that, in all cases, the LePa loss results in the best generalization to out-of-domain task sets and novel label sets, despite not achieving the best generalization on the source distribution. These results suggest that LePa can help to learn prompt parameters that are more generalizable for image classification. The authors also perform some analysis of their method, including ablating over the hyperparameters of each loss term, over the length and depth of the learnable embeddings, the number of worst-case class prompts used, and the underlying ViT instances.

**Questions:**

Could you clarify what this sentence means? I have trouble following it (line 95-97): "performance growth is not significant through the visual embeddings learning directly based on textual embeddings learning."

Additionally, if you could address my issues around the motivation and clarity of the decisions made in constructing LePa that would be great!

Finally, is there any empirical evidence from your experiments that the LePa loss actually serves to improve the balance between the Visual and CE losses throughout training?

**Ethical Concerns:**

["NO or VERY MINOR ethics concerns only"]

**Limitations:**

I think they discuss the limitations of their proposed method well, but there is not discussion on the limitations of how they analyze the proposed method.

**Quality:**

3

**Strengths And Weaknesses:**

Summary of Strengths


The empirical results of the proposed LePa loss are quite extensive and all show pretty convincingly that LePa is more effective than the base tuning methods, as well as the DePT plugin, at preserving model generalization to unseen datasets and classes during image classification prompt-tuning. Moreover, the method is extremely simple to implement and plugin, and it is highly modular, as it appears to be applicable to a number of popular CLIP-tuning methods.

Summary of Weaknesses


Although the method's results are strong, the motivation for various components of the method are not well-explained or motivated in the paper. The method makes several key decisions in computing the LePa loss that are not empirically validated or even, as far as I can tell, well motivated in the paper. For instance, LePa uses the HCE loss which is computed using text representations from the class prompts without input from the tuned parameters. It is not clear to me why this decision is made, or is necessary. The LePa loss is constructed using the HCE loss and the Visual regularization loss but the textual regularization loss is ignored; this is fine if the problem in CLIP tuning is always that there is an imbalance between the CE loss and the Visual loss, and the Textual loss is optimized well each time but there is not empirical evidence given to suggest this nor is it shown that the method actually balances these losses as expected during training. Finally, while the decision to only consider the top-k empirically worse class prompts is validated in the ablations, the motivation for doing this to begin with is, in my opinion, not stated clearly during the introduction of the method.

The point about comparing the rate at which losses are optimized is especially significant to me. L1 and Cross-Entropy losses are not necessarily on the same scale, and it is not clear to me that taking the L1 loss between them is a principled way to ensure that the losses get optimized in a balanced manner. I think the paper would benefit significantly from showing the results of how the trajectory Visual and CE loss imbalance is changed by adding the LePa loss to the models.

Finally, while the ablation over the number of worst-cases considered is useful, as it helps us to understand the role of aggregating a few worst-case prompts, the ablations over e.g. model size, embedding size, etc. don't make much sense as stand-alone results. Suggesting that the method is effective on different architectures doesn't make sense without the context of the performance of those architectures without the LePa loss applied. Similarly, it is not clear whether the ablations over hyper-parameters will transfer to other tasks, e.g. if I was going to use the LePa method for my own task I would likely still need to tune these parameters myself. So their purpose in the paper isn't clear to me.

---

> ### Author Rebuttal · Authors · 2025-07-29
>
> # ***Thanks to the reviewer:***
>
>
> > We would like to express our gratitude to the reviewers: 1) providing rich and comprehensive comments for our work, 2) investing a significant amount of time and effort in reviewing our work, and 3) providing practical guidances, which has greatly improved our work. We will **carefully revise step by step according to opinions of reviewer, and express the respect for your time**.
>
> ***
>
> # ***Response:***
>
> >**Q1**:  Could you clarify what this sentence (line 95-97) means? "performance growth is not significant through the visual embeddings learning directly based on textual embeddings learning."
>
> **A1**: Thank you very much to the reviewer for important comments. We promise to make careful revisions.
>
> In this sentence, we mean, "the visual embeddings learning is directly based on textual embeddings learning" refers to VLP architecture. This VLP architecture directly builds the visual prompt learning structure on the text prompt learning structure. The generalization performance of this VLP architecture is generally not very good, as shown in Tables 1 and 2. We **add experiments** for cross-dataset generalization and domain generalization on the single branch methods of KgCoOp [1] and TCP [2]. We found that the structure of dual branch VLP learning did not perform much better than that of single branch learning in cross-dataset experiments and domain generalization experiments, only slightly higher. This shows that the VLP architecture has no significant improvement in generalization tasks compared to the single-branch text learning architecture.
>
>
> * Domain generalization
> | Method | ImageNet (Training) | -V2| -S | -A | -R | Average |
> |:------ |:------:|:-------:|:------:|:------:|:-------:|:------:|
> | **KgCoOp**  | 70.70 | 64.30 |48.90 | 50.35 | 77.35 | **60.23** |
> | **TCP**  | 71.40 | 64.05 |49.10 | 50.55 | 77.80 | **60.37** |
> | MaPLe| 70.72 | 64.07| 49.15 | 50.90 | 76.98| 60.27 |
> | PromptSRC| 71.27 |64.35| 49.55| 50.90 |77.80 |60.65|
> |  TAP  | 72.50| 64.10 |49.15 |51.30| 77.50 |60.51|
>
>
> * Cross-dataset generalization
> | Method| ImageNet (Training) | Average |
> |:------|:------:|:------:|
> | **KgCoOp**  | 70.70 |  **65.95** |
> | **TCP**   | 71.40 |  **66.29** |
> | MaPLe | 70.72 | 66.30|
> | PromptSRC| 71.27 |65.81|
> |  TAP| 72.50 | 66.02|
>
>
> >**Q2**:   The motivation and clarity of the LePa.
>
> **A2**: Thank you very much to the reviewer for important comments. We will add more descriptions and explanations in future versions, and we **promise** to revise carefully.
>
> * The input of this LePa method is HCE, which is composed of handwritten prompt words. Why do we use handwritten prompts? It is because the pure CLIP (ICML2021) architecture has relatively good generalization performance, which is due [1,2,3,6,7] to the use of numerous handwritten prompt word templates in CLIP. So we thought of exploring the natural semantics of these prompt words of CLIP official library and analyzing relatively poor examples. The LePa loss calculates the absolute value of the HCE loss and visual loss after subtraction, in order to constrain the convergence of these two losses as consistent as possible during the training process.
>
> * The relationship between generalization and multi-loss uncoordinated learning has been experimentally and empirically demonstrated in literature [3, 6, 7]. The theoretical basis is that the purpose of CE loss is to fine tune and enhance few-shot ability [9], while the purpose of visual loss is to avoid forgetting pre-trained features and enhance generalization ability [3,6,7]. There is a compromise and balance between these two losses, a natural conflict where one goes up and the other goes down [8]. To prevent visual loss convergence instability and affect CE loss, we propose an HCE that normalizes HCE and visual loss, using this method to maintain the stability of visual loss. HCE is composed of pre-trained manual text features and has generalization ability [1,2,3,6,7].
>
> * Phenomenon of trajectory: In the convergence process of visual loss and CE loss, there will be fluctuations in visual loss, specifically manifested as: At the beginning of training, visual loss is sometimes very small, and the CE loss is relatively large at this time. At the end of training, the CE loss converges quickly and relatively small, and the visual loss is sometimes relatively large and converges slowly. The convergence of CE loss is stable, and the convergence of visual loss is unstable.  Now, through our method, both visual loss and CE loss have steadily decreased. We have alleviated this phenomenon of uncoordinated learning.
>
> * Our purpose of analyzing relatively poor cases is to increase robustness. These methods [1,2,3,6,7] have a very important motivation and explanation, which is that handwritten prompt words have strong generalization through pre-training CLIP generated features. So HCE has a specific function, which is to obtain cross entropy loss with generalization.
>
> * Our method does not use the loss of text branching. In Figure 2, we provide an example based on PromptSRC. The PromptSRC method requires Loss-textual, but there are still many methods that do not have Loss-textual, such as PLOT, MaPLe, etc. Our method requires visual loss. Text loss is not necessary for us, it may exist in baseline work.
>
> * In the paper, based on PromptSRC, the embedding length is 4 and the embedding depth is 9. Our method, as a plugin, does not change the hyperparameter settings for embedding in baseline work. Therefore, in our paper, based on the experimental performance generated by other people's work, the hyperparameters in terms of embeddings are all settings written in other papers. The reason why we conducted this ablation experiment is to verify whether our plugin can still perform without changing the embedding settings of other baselines.
>
>
>
>
>
>
> >**Q3**:  Is there any empirical evidence from your experiments that the LePa loss actually serves to improve the balance between the Visual and CE losses throughout training?
>
> **A3**: Thank you very much to the reviewer for important comments.
>
> * Phenomenon of evidence: In the convergence process of visual loss and CE loss, there will be fluctuations in visual loss, specifically manifested as: At the beginning of training, visual loss is sometimes very small, and the CE loss is relatively large at this time. At the end of training, the CE loss converges quickly and relatively small, and the visual loss is sometimes large and converges slowly. The convergence of CE loss is stable, and the convergence of visual loss is unstable.  Now, through our method, both visual loss and CE loss have steadily decreased in our epochs of settings . Visual loss is constrained by HCE and decreases smoothly. We have alleviated this phenomenon of uncoordinated learning.
>
>
>
> ***
>
> [1]Visual language prompt tuning with knowledge-guided context optimization
>
> [2]Textual based class-aware prompt tuning for visual-language model
>
> [3]Consistent prompt learning for vision-language models
>
> [4]Tree of attributes prompt learning for vision-language models
>
> [5]Tuning multi-mode token-level prompt alignment across modalities
>
> [6]Consistency-guided prompt learning for vision-language models
>
> [7]Self-regulating prompts: Foundational model adaptation without forgetting
>
> [8]Dept: Decoupled prompt tuning
>
> [9]Learning to prompt for vision-language models

---

> > ### Comment · Area_Chair_95bD · 2025-08-05
> >
> > Dear Reviewer Eibv,
> >
> > Please respond to authors’ rebuttals and participate in discussions. Thanks.
> >
> > Best regards,
> > AC

---

> ### Author Response · Authors · 2025-08-04
> **Window for discussion is closing**
>
> Dear Reviewer,
>
> Thanks a lot for your time in reviewing and insightful comments, according to which we have carefully revised the paper to answer the questions. We sincerely understand you’re busy. But since the discussion due is approaching, would you mind checking the response to confirm where you have any further questions?
>
> We are looking forward to your reply and happy to answer your further questions.
>
> Best regards
>
> Authors

---

> ### Author Response · Authors · 2025-08-06
> **Would you mind raising the score**
>
> Dear reviewer,
>
> Your constructive comments have greatly helped us improve our paper. Do you have any other concerns right now? If you have no further questions/concerns, would you mind raising the score? Your evaluation of our work is invaluable and we greatly appreciate your time.
>
> Best regards and thanks,
>
> Authors

---

> ### Author Response · Authors · 2025-08-07
> **Window for discussion is closing**
>
> Dear Reviewer,
>
> Thanks a lot for your time in reviewing and insightful comments, according to which we have carefully revised the paper to answer the questions. We sincerely understand you’re busy. But since the discussion due is approaching, would you mind checking the response to confirm where you have any further questions?
>
> We are looking forward to your reply and happy to answer your further questions.
>
> Best regards
>
> Authors

---

> ### Author Response · Authors · 2025-08-08
> **Could you please consider improving score?**
>
> Dear reviewer,
>
> Your constructive comments have greatly helped us improve our paper. Do you have any other concerns right now? If you have no further questions/concerns, would you mind raising the score? Your evaluation of our work is invaluable.
>
> Best regards and thanks,
>
> Authors

---

### Note · Authors · 2025-08-12

*We would like to thank reviewer **osmA** for the substantial **score** increase and **encouragement** for our work. We thank the other reviewers for their technical review of our work and the **time they spent**. We thank **AC** for facilitating communication between reviewers and authors.*

*We would like to emphasize the highlights of our work:*

***

***1) Classic works***

* Our research area has been frequently featured in ICCV, CVPR, ICML, and ICLR.

***2) Topic matching***

* We have learned that our topic (prompt learning + CLIP) is very compatible with the topic of NeurIPS.

***3) Practicality***

* Our task is to collect 11 real-world datasets, and our work has practical application value and will have value for human society.

* All reviewers acknowledged our rich experiments.

***4) Novelty***

* In the weakness, all reviewers have no question about the novelty. This reflects the novelty of our work.

***5) Design***

* All reviewers chose the Strengths option and recognized that our design is plug-and-play.

***6) Promise***

* We will do our best to solve all the questions raised by the reviewers, and provide AC and reviewers with a satisfactory PDF file before the deadline to thank you for your time and effort.

***7) Contributing to community***

* We sincerely hope that our work can contribute to the **NeurIPS technology community.**

---

### Decision · Program_Chairs · 2025-09-17

**Decision:**

Accept (poster)

**Comment:**

This paper presents a plug-in module, called Levelling Paradigm (LePa), to improve the generalization of Vision-Language Prompting (VLP) frameworks, in application settings without labeled data. LePa tackles the challenge of uncoordinated optimization between vision-language alignment and visual feature preservation by incorporating a dynamic constraint objective. It integrates seamlessly with various VLP baselines and demonstrates consistent performance gains across 11 datasets. While LePa mitigates uncoordinated learning between components, the new loss may itself introduce optimization conflicts with existing objectives, which is considered a weakness.